# New Therapeutics for Heart Failure: Focusing on cGMP Signaling

**DOI:** 10.3390/ijms241612866

**Published:** 2023-08-16

**Authors:** Supachoke Mangmool, Ratchanee Duangrat, Warisara Parichatikanond, Hitoshi Kurose

**Affiliations:** 1Department of Pharmacology, Faculty of Science, Mahidol University, Bangkok 10400, Thailand; supachoke.man@mahidol.ac.th (S.M.); ratchanee.dun@student.mahidol.ac.th (R.D.); 2Department of Pharmacology, Faculty of Pharmacy, Mahidol University, Bangkok 10400, Thailand; warisara.par@mahidol.ac.th; 3Pharmacology for Life Sciences, Graduate School of Pharmaceutical Sciences, Tokushima University, Tokushima 770-8505, Japan

**Keywords:** cyclic guanosine monophosphate (cGMP) signaling, natriuretic peptides, soluble guanylyl cyclase (sGC), heart failure (HF), drugs

## Abstract

Current drugs for treating heart failure (HF), for example, angiotensin II receptor blockers and β-blockers, possess specific target molecules involved in the regulation of the cardiac circulatory system. However, most clinically approved drugs are effective in the treatment of HF with reduced ejection fraction (HFrEF). Novel drug classes, including angiotensin receptor blocker/neprilysin inhibitor (ARNI), sodium-glucose co-transporter-2 (SGLT2) inhibitor, hyperpolarization-activated cyclic nucleotide-gated (HCN) channel blocker, soluble guanylyl cyclase (sGC) stimulator/activator, and cardiac myosin activator, have recently been introduced for HF intervention based on their proposed novel mechanisms. SGLT2 inhibitors have been shown to be effective not only for HFrEF but also for HF with preserved ejection fraction (HFpEF). In the myocardium, excess cyclic adenosine monophosphate (cAMP) stimulation has detrimental effects on HFrEF, whereas cyclic guanosine monophosphate (cGMP) signaling inhibits cAMP-mediated responses. Thus, molecules participating in cGMP signaling are promising targets of novel drugs for HF. In this review, we summarize molecular pathways of cGMP signaling and clinical trials of emerging drug classes targeting cGMP signaling in the treatment of HF.

## 1. Introduction

Heart failure (HF) is a condition where the heart’s ability to pump and deliver sufficient oxygen and nutrients to peripheral tissues is impaired. HF can be divided into two major categories based on the ejection fraction (EF) of the left ventricular (LV): HF with reduced EF (HFrEF; EF of 35–40% or less) and HF with preserved EF (HFpEF; EF of 50% or more), in which the EF is preserved but LV diastolic function is reduced. HFrEF and HFpEF patients account for 30–50% and 50%, respectively. Currently, most clinical drugs for HF are effective in treating HFrEF but not HFpEF [1].

The current standard treatment for HFrEF includes many drug classes that are angiotensin-converting enzyme (ACE) inhibitors or angiotensin II receptor blockers (ARBs) (if ACE inhibitors are not tolerated), β-blockers, and mineralocorticoid receptor blockers [1]. However, drugs with novel mechanisms of action have been developed and are now being used in clinical practice; these include angiotensin receptor blocker/neprilysin inhibitor (ARNI), sodium-glucose co-transporter-2 (SGLT2) inhibitor, hyperpolarization-activated cyclic nucleotide-gated (HCN) channel blocker, soluble guanylyl cyclase (sGC) stimulator/activator, and cardiac myosin activator.

Cyclic guanosine monophosphate (cGMP) acts as a crucial second messenger, as do cyclic adenosine monophosphate (cAMP) and Ca^2+^. cGMP, produced by guanylyl cyclase (GC), triggers responses by regulating the activities of downstream signaling molecules. cGMP signaling represents molecules participating in the pathway starting from the production of cGMP to the regulation of the molecule’s activity [2]. cGMP signaling has received considerable attention because of its involvement in the treatment of HF. Mechanistic analysis has revealed that cGMP signaling acts to prevent pathological remodeling, and large clinical trials support the importance of cGMP signaling [1]. Numerous drug classes, including ARNIs, SGLT2 inhibitors, and sGC activators introduced for HF management are directly or indirectly associated with cGMP signaling. Although cGMP is not involved in the action of ivabradine, cGMP can bind its target molecule [2,3].

When the four drug classes—β-blockers, mineralocorticoid receptor antagonists, ARNIs, and SGLT2 inhibitors—are administered early and appropriately to HF patients, they can increase survival rate and reduce hospitalizations due to HF. Thus, these drug classes are referred to as the “fantastic four”, meaning the main drug classes proposed as the central regulators of future HF regimens [4]. cGMP signaling is terminated by the hydrolysis of cGMP by phosphodiesterases (PDEs) or via export to the extracellular space through multidrug resistance proteins known as ABC transporters [2]. The signaling molecules that link cGMP signaling to cellular responses are varied from cell to cell. Furthermore, the expression of intracellular signaling molecules differs between healthy and diseased conditions [2,3]. In this review, we introduce cGMP signaling molecules and then extensively discuss their roles as potential therapeutic targets for HF therapy. Published literature reporting evidence of novel drug classes for HF acting via cGMP signaling was comprehensively searched for in standard electronic databases such as PubMed, Embase, ScienceDirect, and Scopus.

## 2. Cyclic Guanosine Monophosphate (cGMP) Signaling

There are three aspects of cGMP signaling: production, effectors, and elimination from the cells. Elimination includes degradation and export via transporters [3].

### 2.1. cGMP Production

cGMP is produced by soluble guanylyl cyclase (sGC) and membrane-bound guanylyl cyclase (Table 1) [3]. sGC is present in the cytoplasm and is characterized as a heterodimer composed of α and β subunits—sGC1 (α1β1) and sGC2 (α2β1) isoforms. sGC contains heme and Fe^2+^. When Fe^2+^ is oxidized to Fe^3+^, it becomes an inactive apo-protein that does not respond to nitric oxide (NO) [3].

GC-C, GC-D, GC-E, GC-F, and GC-G are not associated with HF because they are pseudogenes whose expression is detectable in a few tissues. sGC stimulators increase their activity only when sGC is in Fe^2+^ form. In contrast, sGC activators act on oxidized sGC (Fe^3+^-sGC or Apo type sGC) and increase sGC activity [5,6,7]. The list of sGC stimulators and activators is shown in Table 2.

Membrane-bound GCs are molecules with extracellular amino termini, transmembrane regions, and intracellular regions. The intracellular region has an ATP binding site and a catalytic domain. The binding of ATP to membrane-bound GC potentiates GC activity [2,3]. There are seven isoforms of membrane-bound GCs: from GC-A to GC-G. GC-A (called natriuretic peptide receptor-A, NPR-A, or NPR1) and GC-B (called NPR-B or NPR2) are the most extensively analyzed. The ligands that activate NPR-A and -B are the natriuretic peptides (NPs), atrial NP (ANP) and brain NP (BNP). GC-D and GC-G are not present in humans. BNP has cardiovascular (CV) protective effects as well as metabolic effects such as promoting lipolysis and improving insulin resistance [6,7]. The heart is the main organ that produces ANP and BNP. ANP is produced by the atrium while BNP is produced by the ventricle. The production of these peptides in organs other than the heart is exclusively limited in humans. It is reasonable that almost 100% of the ANP and BNP circulating in the blood originate from the heart. BNP and N-terminal fragments of the BNP precursor (NT-proBNP) can be used as biomarkers of HF. BNP and NT-proBNP levels increase during the development of HF but decrease as the drug treatment becomes apparent [1].

### 2.2. cGMP Signaling Effector Molecules

cGMP evokes physiological responses by binding to three types of effector molecules: cGMP-dependent protein kinases (PKGs), PDEs, and cyclic nucleotide-activated cation channels [6,7]. Cyclic nucleotide-activated cation channels comprise two families, cyclic nucleotide-gated (CNG) channels and hyperpolarization-activated cyclic nucleotide-gated (HCN) channels, that directly couple the activation of the channel by nucleotides with the influx of ions. CNG channels are opened by binding to cAMP or cGMP, whereas HCN channels are mainly regulated by voltage. HCN channels are regulated by cyclic nucleotide as well as voltage, and cyclic nucleotide binding to HCN channels increases their probability of opening. HCN channels are specifically regulated by cAMP. The guanine nucleotide exchange factor that is regulated by cGMP has been identified. Production and mechanisms of action of cGMP are presented in Figure 1.

#### 2.2.1. cGMP-Dependent Protein Kinases (PKGs)

cGMP directly activates several target molecules, including PKGs, PDEs, and ion channels. Of these, PKG is the most well-characterized signaling molecule that is activated by cGMP. PKG is classified into two families: PKGI and PKGII. PKGI has two isoforms: PKGIα and PKGIβ, these two isoforms differ in the N-terminal leucine zipper (LZ) domain [8]. The LZ domain at the amino terminus is used for substrate binding and dimer formation between isoforms [8]. In smooth muscle cells, oxidative modification of the cysteine at position 42 in PKG promotes disulfide bonding between dimers, which in turn promotes the translocation of the oxidized PKG to the plasma membrane [9]. On the plasma membrane, PKG phosphorylates Ca^2+^-activated potassium channels (BK_Ca_) and causes membrane hyperpolarization, resulting in vascular relaxation and hypotension. Since PKG induces vasodilation, the use of drugs that activate PKG is largely limited due to hypotension [9].

Several target proteins, including transient receptor potential cation channel, subfamily C, member 6 (TRPC6), regulator of G-protein signaling 2 and 4 (RGS2 and RGS4), sGC, phospholamban, and cardiac myosin binding protein-C (cMyBP-C) have been reported as substrates for PKG [9]. TRPC6 channels are voltage-independent cation channels and are activated by diacylglycerol (DAG). Upon receptor stimulation, Gq induces phospholipase C (PLC) activity and liberates inositol-1,4,5-trisphosphate (IP_3_) and DAG. IP_3_ binds to the IP_3_ receptor and promotes the release of Ca^2+^ from intracellular Ca^2+^ stores, while DAG activates protein kinase C as well as TRPC3 and TRPC6 channels to translocate cations such as Ca^2+^ into the cells. TRPC3/6 channels are signaling molecules that mediate the receptor-stimulated cardiac hypertrophic response and pressure overload-induced hypertrophy [10]. PKG phosphorylation of TRPC6 attenuates channel activity and suppresses pressure overload-induced hypertrophy [11]. PKG phosphorylates Ser64 at the α1 subunit of sGC, thereby inhibiting its activity. Thus, PKG catalyzes the phosphorylation of sGC and forms negative feedback [12]. cMyBP-C, a cardiomyocyte-specific protein, has been shown to bind PKGIα, which phosphorylates amino acids at three specific sites and inhibits pathological remodeling [13]. Thus, PKG reduces afterload via vasodilation and cardiac remodeling via phosphorylation of cMyBP-C.

The role of PKG in the heart is complicated. However, cardiac-specific PKG knockout mice and LZ domain-mutated PKGIα knock-in mice that retain the kinase activity but cannot bind substrate through the LZ domain are currently available [14]. Systemic knockout of PKG caused modest increases in blood pressure (BP) and the development of progressive hypertrophy [14]. LZ domain-mutated mice developed adult-onset hypertension and abnormalities of vascular relaxation [14]. Increased hypertrophy and remodeling were observed in pressure-overloaded mutant mice; this eventually progressed to markedly severe HF [15]. Thus, PKGIα protects the heart against stress-induced responses [9]. The LZ-domain-mutated knock-in mice and cardiac myocyte-restricted deletion mice will reveal the physiological functions and target molecules of cardiac PKGIα in more detail.

ANP-bound GC-A increases cGMP, which antagonizes the Ca^2+^-dependent hypertrophic response via angiotensin II (Ang II) stimulation at angiotensin II type I receptor (AT_1_R). Ang II–AT_1_R–Gq signaling is inhibited by ANP stimulation and increases cGMP and PKG activation. PKG phosphorylates RGS4 and promotes the formation of a complex between RGS4 and Gαq, causing attenuation of Gαq activation [16]. RGS proteins are a family of 21 members [17]. Each RGS protein selectively binds α subunits of G protein; Gs, Gi, Gq, and G12. RGS2-mediated Gαq inhibition by PKG-catalyzed phosphorylation has also been reported [16,18,19]. Thus, the phosphorylation of RGS2 by PKGI inhibits Gq signaling and suppresses TRPC3/TRPC6 channel activity. The binding of RGS2 and RGS4 to Gαq is negative feedback regulated by PKG [16,18,19].

Studies suggest that the differential PKG pools of sGC and membrane-bound GC are critical [20]. PKG activated by NO/sGC/cGMP signaling phosphorylates several myofilament proteins to enhance the relaxation response and attenuate the contractility-enhancing effects of β-adrenergic receptors [21,22,23]. In contrast, PKG activation by membrane-bound GC/cGMP signaling selectively enhances the activity of transcription factors associated with enhanced cell survival and adaptation, GATA binding protein 4 (GATA4), and cAMP response element binding protein (CREB) [20]. Reactive oxygen species (ROS) attenuate PKG activation and protective signaling in cardiomyocytes [24]. PKG inhibits activation and nuclear trafficking of mothers against decapentaplegic homolog 3 (SMAD3), thus transforming growth factor-beta (TGF-β)-related signaling [25,26]. Another redox-dependent modification of PKG1α is S-guanylation at Cys195, which represents a covalent modification of cGMP [27]. Under excess ROS and NO, cGMP is converted to 8-nitro-cGMP, which reacts with the cysteine residues of PKG1α to generate the S-guanylated form. S-guanylated PKG1α is persistently active. However, the contribution of redox/NO-dependent modification of PKG1α to the development of HF is unknown.

#### 2.2.2. Phosphodiesterases (PDEs)

PDEs consist of at least 11 members whose activities are modulated by various factors (Table 3). PDE1 activity is increased by Ca^2+^/calmodulin. cGMP binds to PDE2 and promotes cAMP degradation. In human atrial myocytes, compounds that activate sGC and increase cGMP levels enhance L-type calcium current via PDE3 inhibition [28]. PDE3 is selective for cAMP but also binds cGMP. cGMP binding inhibits cAMP binding and increases cAMP-mediated actions. PDE3 is also phosphorylated by protein kinase A (PKA), which increases its cAMP-hydrolyzing activity. PDE5/6/9 are cGMP-selective PDEs [29,30].

PDE9, a cGMP-selective PDE, has been reported to be selective for cGMPs to be degraded; PDE9A selectively degrades cGMPs generated by natriuretic peptide (NP)–NP receptors (pGC) rather than cGMPs generated by NO-sGC [31]. Genetic or pharmacological inhibition of PDE9A improved cardiac function in pressure overload mice undergoing transverse aortic constriction [31], suggesting that PDE9A inhibition is a new potential target for HF therapy. However, the outcome obtained from a pressure overload model is proposed to be equivalent to a therapeutic agent for HFrEF but not for HFpEF.

Studies on myocardial targets of PDE9-related cGMP–PKG signaling are still limited. Most previous studies have not strictly distinguished between sGC- and membrane-bound-regulated cGMP signaling. Future studies will reveal the crosstalk between these functional compartments as well as their respective importance in various pathological conditions. This will provide a complete and modified framework of compartmentalized PKG-related signaling.

#### 2.2.3. Cyclic Nucleotide-Regulated Cation Channels

Cyclic nucleotide-regulated cation channels comprise two families: CNG and HCN channels [32]. CNG channels are opened by direct binding of cAMP or cGMP, while HCN channels are activated upon membrane hyperpolarization. The CNG family comprises six members categorized into A subunits (CNGA1–4) and B subunits (CNGB1 and CNGB3). The HCN family consists of four members: HCN1–4. Four subunits assemble to form functional CNG and HCN channels, but there is no report indicating that CNG and HCN channels form heterotetramers. In the cytosolic carboxyl terminus, all subunits carry a cyclic nucleotide-binding domain (CNBD). The binding of cAMP to the CNBD of HCN channels induces conformational changes, resulting in an increased probability of channel opening [32,33].

CNG channels are extensively distributed in the retina and are responsible for photoreception. The affinity of CNG channels is much higher for cGMP than for cAMP [33,34]. Of the CNG channels, only CNG3 channels are expressed in the heart and have been reported to mediate Ca^2+^ elevation when cGMP is increased [35]. However, it has not been demonstrated that CNG3 channels are involved in the development or exacerbation of HF symptoms. The A1–A3 subunits of the CNG family, but not the A4 subunit, can form homomeric channels on their own. On the other hand, the B subunit cannot form a functional channel by itself. When expressed with the A subunit, the B subunit becomes highly sensitive to cAMP, similar to cGMP-sensitive CNG channels [32,33,34]. Although CNG channels are mainly expressed in the retina (rods and cones), some subunits are also expressed in the heart and testis, for instance, CNG3 channels (A3 subunit) [35].

CNG channels are tetrameric voltage-independent cation channels [32]. CNG channels were first discovered in rod photoreceptors, where photoreception by rhodopsin is transmitted to transducin, which then stimulates PDEs and lowers intracellular cGMP. This is followed by the closure of CNG channels and a reduction in ‘dark current’. Similar channels have been detected in the cilia and pineal glands of olfactory neurons [33]. In contrast, HCN channels are cation channels that are activated by hyperpolarization at negative voltages (approximately −50 mV). cAMP and cGMP directly activate the channel, shifting the activation curve of the HCN channels to a more positive voltage [34]. HCN channels are responsible for the pacemaker currents found in many excitable cells, including cardiac cells and neurons. Four known HCN channels have six transmembrane domains and form tetramers. The channels are thought to be able to form heteromers with each other, as has been shown for HCN1 and HCN4 [34].

HCN channels are expressed in the sinoatrial (SA) node and control heart rate (HR). The affinity of cAMP for HCN channels is about 10-fold higher than that of cGMP [33,34]. In the case of HCN channels, increased cAMP binds to CNBD of HCN4, which increases the open state of HCN channels in response to voltage. As the open state of the channels increases, HR increases [33,34]. HCN channels are phosphorylated by PKA. PKA-catalyzed phosphorylation of HCN4 channels increases the firing rate in response to sympathetic stimulation [36].

#### 2.2.4. Nitric Oxide (NO)

sGC is activated by NO; NO not only activates sGC but also exerts its action through direct modification of cysteine residues (S-nitrosylation) on various proteins (see the extensive review of S-nitrosylation of signaling proteins in addition to molecules that are involved in HF [37]).

### 2.3. cGMP Elimination Process

cGMP levels are necessary to be rapidly adjusted to the basal level since excessive cGMP levels activate cellular signaling cascades that have detrimental effects on the cells. In general, cGMP levels are normalized via two pathways: cGMP degradation by PDEs or cGMP exportation by multidrug resistance-associated proteins (MRPs) [38]. cGMP is decreased by MRP-dependent cGMP export [38]. MK571, an inhibitor of MRP4 and MRP5, was applied to human coronary artery smooth muscle cells, mouse aorta, and pressure overload-induced HF mouse models. The results showed that cGMP generated by ANP stimulation was exported via the MK571-mediated pathway with a potent vasorelaxation response [39]. Furthermore, MK571 alone did not alter BP, but the hypotensive effect of ANP was markedly enhanced following the administration of MK571 [39]. Thus, MK571 treatment improved several cardiac functional parameters of HF in mice. These MK571 results may be due to the inhibition of MRP4- and MRP5-mediated export and increased intracellular cGMP levels.

## 3. New Drugs for the Treatment of HF

### 3.1. Angiotensin Receptor Blocker/Neprilysin Inhibitor (ARNI)

ARNI is a single-molecule combination of ARB, valsartan, and the neprilysin inhibitor, sacubitril (activated by esterase and converted to the neprilysin inhibitor LBQ657) (Figure 2). Neprilysin is an enzyme that degrades natriuretic peptides (ANP and BNP), which are secreted by the heart and known to inhibit the renin-angiotensin system and sympathetic nervous system. ANP and BNP are used for the treatment of acute HF through continuous intravenous administration, which inconveniences chronic HF treatment [40,41].

Neprilysin is expressed in the kidneys, vascular smooth muscles, and lungs. Besides ANP and BNP, neprilysin degrades vasodilating hormones such as adrenomedullin and bradykinin, and vasoconstrictive hormones such as endothelin-1 (ET-1) and angiotensin II (Ang II) [40,41]. Therefore, inhibiting neprilysin alone increases cardioprotective factors (ANP, BNP, etc.) as well as cardiac stress factors (ET-1 and Ang II). ARNI is a drug class that enhances the protective effect by inhibiting Ang II stimulation with valsartan while also inhibiting neprilysin, which potentiates the effects of ANP and BNP [40,41].

### 3.2. Sodium-Glucose Co-Transporter-2 (SGLT2) Inhibitors

Glucose transporters are classified into two main families: glucose transporters (GLUTs) and sodium-glucose cotransporters (SGLTs) [42]. GLUTs translocate extracellular glucose into the cell along a glucose concentration gradient via facilitated diffusion, while SGLT is a sodium-dependent glucose cotransporter that translocates glucose using the difference between intracellular and extracellular sodium concentrations as a driving force [42].

Currently, the GLUT family has 14 members (Table 4) [43]. GLUT4, GLUT8, and GLUT12 are responsible for insulin-stimulated glucose uptake by promoting the translocation of GLUTs to the plasma membrane. In the heart, glucose uptake by GLUT4 is important due to the higher expression of GLUT4 compared with other isoforms. Glucose taken up by cells is metabolized to glucose-6-phosphate, which is used in the TCA pathway (mitochondrial ATP production), polyol pathway, and pentose phosphate pathway (conversion of glycogen) [43].

The SGLT family encompasses seven isoforms with different physiological functions (Table 5). SGLT2 is expressed in the proximal tubules of the kidneys and is responsible for approximately 90% of glucose reabsorption. SGLT2 inhibitors are widely used to treat diabetes by promoting urinary glucose excretion. However, SGLT2 inhibitors are also effective in the treatment of HF. In a clinical trial of diabetic patients, empagliflozin—an SGLT2 inhibitor—prevented hospitalization with HF symptoms and death due to CV causes [44,45]. Subsequent clinical trials demonstrated that empagliflozin and dapagliflozin also reduced HF and death due to CV causes in patients without diabetes. The effects of SGLT2 inhibitors on the reduction of death and hospitalization were almost equal in diabetes and non-diabetes patients [46]. These studies revealed that SGLT2 inhibitors are effective in treating HF regardless of diabetes. They also demonstrated that the effects of SGLT2 Inhibitors are not secondary to the improvement of diabetes.

Although the myocardium also expresses SGLT1 and sodium myoinositol cotransporter 1 (SMIT1), their activities are not affected by SGLT2 inhibitors [42]. Thus, the underlying mechanisms through which renal responses by SGLT2 inhibitors are transmitted to the heart are of interest. SGLT2 inhibitors inhibit sodium reabsorption and promote natriuresis. Although natriuresis is a protective response against the CV system, the diuretic effect is a transient response observed early during treatment [1]. Thus, the diuretic effect alone cannot explain the long-term effects on HF. To date, no single mechanism has been able to encompass the effects of SGLT2 inhibitors on HF. Improvement of the ionic environment within the myocardium through anti-inflammatory effects and inhibition of the Na^+^/H^+^ exchange transporter (NHE3), whose activity is increased in HF, has been implicated in cardiac tissue damage [47]. Improvement of mitochondrial function also contributes to the protective effects of SGLT2 inhibitors on the kidney. However, it is believed that the cardioprotective effects are explained by a combination of various effects via improved energy metabolism [48,49].

The selectivity of SGLT2 inhibitors for SGLT2 over SGLT1 varies from 290-fold for canagliflozin to 2900-fold for tofogliflozin [50]. Currently, ARBs and β-blockers are effective against HFrEF but not HFpEF. Remarkably, SGLT2 inhibitors are effective at treating both HFpEF and HFrEF [51]. Analysis of SGLT2 inhibitors against HFpEF is expected to lead to the development of a new treatment for HFpEF.

### 3.3. Hyperpolarization-Activated Cyclic Nucleotide-Gated (HCN) Channel Blocker

HCN4 is mainly involved in slow depolarization during the action potential occurring in the SA node to control the HR [34]. Ivabradine selectively inhibits HCN channels, including HCN4. In cells expressing HCN4, ivabradine inhibited HCN4 activity at IC_50_ of about 0.4 μM. Furthermore, it also inhibited L-type Ca^2+^ channels and Na^+^ channels [52]. Regarding selectivity, ivabradine inhibits HCN1 channels within a concentration range similar to that of HCN4. Thus, when ivabradine acts on HCN4, HCN1 is also inhibited [52]. In the SA node, ivabradine slows depolarization by inhibiting HCN4. However, ivabradine does not alter cardiac contractility [53].

In patients with HFrEF, the risk of death or hospitalization increases when the resting HR exceeds 70 beats/min [1]. Therefore, ivabradine is recommended when standard medications are ineffective at decreasing HR. At the SA node, an action potential is generated by spontaneous and gradual depolarization of membrane potential called the pacemaker current (*I*_f_), which is triggered by the influx of cations through HCN4 channels [54]. Ivabradine selectively blocks HCN4 channels in the SA node, leading to the inhibition of Na^+^ influx through HCN4 channels. As a result, the slope of phase IV depolarization becomes slower and the HR decreases. Because ivabradine acts on HCN4 channels only in the SA node, it reduces HR without affecting cardiac functions such as conduction, contractility, or repolarization of the heart [53,54]. In patients with HFrEF, a high HR is associated with a risk of death or hospitalization, and ivabradine has been reported to reduce the high HR-associated risk [1].

Cardiac-specific HCN4-channel knockout mice showed slower conduction of excitation through the atrioventricular (AV) node [55], suggesting that the HCN4 channels are also involved in conduction through the AV node. The cardiac voltage-gated Na^+^ channel (Nav1.5) is also inhibited by ivabradine [56]. Na^+^ channels in the nervous system (Nav1.2) and skeletal muscles (Nav1.4) are also inhibited within a similar concentration range (low μM) [56]. Thus, ivabradine may inhibit AV node conduction by blocking both HCN channels and voltage-gated Na^+^ channels.

### 3.4. Cardiac Myosin Activators

Inotropic agents have been reported to worsen life expectancy in patients with HFrEF. However, in patients with severe HF, inotropic agents may be necessary for maintaining the circulatory system and sustaining life [1]. Thus, the development of inotropic agents that improve life expectancy and hemodynamics without potential adverse effects is in high demand. Omecamtiv is a myocardial myosin activator that enhances actin–myosin coupling, which is the final step in myocardial contraction. Omecamtiv enhances contractility without increasing Ca^2+^ influx or Ca^2+^ sensitivity to myocytes and therefore does not increase myocardial oxygen consumption [57,58].

Since it does not act directly or indirectly via NO–cGMP signaling, the mechanism of action is briefly presented here. In cardiac muscle, actin binds to the Z-body and induces contractility by myosin sliding between actin fibers. A series of reactions occur between myosin and ATP during the contraction–relaxation cycle. ATP binds to myosin unoccupied by ADP. ATP is then hydrolyzed to ADP and inorganic phosphorus is liberated. ADP is then dissociated from the binding site. Myosin returns to the nucleotide-free state and binds strongly to actin from the inorganic phosphorus-released state to the ADP-dissociated state [59]. Omecamtiv binds to myosin and promotes the dissociation of inorganic phosphorus and inhibits binding of ATP to myosin. Thus, Omecamtiv enhances contractility by increasing the strong binding state of actin and myosin, which is essential for cardiac contractility [60].

### 3.5. Soluble Guanylyl Cyclase (sGC) Stimulator/Activator

The activity of NO-sGC-cGMP signaling is decreased in the hearts of HF patients, resulting in decreased activity of PKG, a target of cGMP. Since PKG has inhibitory effects on myocardial hypertrophy and vasodilation, decreased PKG activity is associated with the development or worsening of HF [61]. Riociguat is an sGC stimulator that stabilizes the binding of NO to sGC and increases the sensitivity of sGC to NO. It stimulates sGC directly in a NO-independent manner [62]. Vericiguat, which has similar activity, but a longer duration of action compared with riociguat, improved HF symptoms in patients and reduced death due to CV defects, with additive effects in patients receiving standard treatment for chronic HF [63].

In patients with HF, NO production and NO-dependent responses are impaired, resulting in a failure of sGC stimulation. Decreased sGC activity leads to vasoconstriction and impaired cardiac function [64]. Belisiguat is a novel sGC stimulator used in HF therapy that targets sGC in the NO–sGC–cGMP signaling pathway. The stimulation of vascular and cardiac sGCs by belisiguat improves vasodilation and cardiac function. It activates the NO–sGC–cGMP pathway through two mechanisms, direct stimulation of sGCs and increased sensitivity of sGCs to endogenous NO, leading to cGMP production [5]. The progression of chronic HF is inhibited when blood vessels dilate and BP is reduced. However, sGC stimulants are also effective at correcting reductions in BP, indicating that their direct action on the heart is also protective against HF [62,63,64]. In other words, a decrease in NO production by endothelial cells results in a decrease in tissue cGMP levels.

In patients with chronic HF, endothelial function is impaired by oxidative stress and inflammation. It has been shown that reduced endothelial function results in reduced NO production, which in turn results in defective sGC activity [62]. cGMP is a signaling molecule that regulates physiological processes such as myocardial contraction, vascular tone, and cardiac remodeling [2,3]. Therefore, a decrease in cGMP levels creates negative feedback between myocardial and vascular dysfunction, which in turn leads to further worsening of HF [62,63,64]. β-Blockers and other drugs that have become the standard of care for HF are not thought to act on the NO–sGC–cGMP signaling pathway. Therefore, novel agents that can activate the NO–sGC–cGMP pathway and increase cGMP are needed to further reduce the risk to HF patients receiving standard therapy [64]. Long-term stimulation of sGC in mice should be conducted with caution, as long-term PKGI activation has been reported to have detrimental effects on the heart, especially in the presence of pressure loading and neurohumoral stress [65].

Clinical trials using drugs targeting PKGI in patients with HF have shown varied efficacies across trials. In patients with HFrEF, PDE5 inhibitors, and the sGC activator vericiguat showed effective endpoints [2,66,67]. However, not all trials achieved the desirable positive outcomes [68,69]. NO is cGMP-independent, making it difficult to assess the contribution of PKG activation on the cardioprotective effects observed using the combination of hydralazine/isosorbide nitrate (a NO-producing agent) and ARNI (valsartan/sacubitril) because neprilysin inhibitors also act in a natriuretic peptide-independent manner and inhibit the degradation of non-natriuretic peptides [70]. The disappointing results of drugs that modulate cGMP signaling in HFpEF patients may also be due to the fact that the elevation of cGMP levels in the heart may be too great or PKGI activation may be too high to be harmful, and pharmacological activation of PKG in HF may have a narrow therapeutic window [2].

### 3.6. SGLT2 Inhibitors and cGMP Signaling

Dapagliflozin has been suggested to have a positive effect on cardiac function and metabolism by enhancing BNP bioactivity and reducing cardiac load. Empagliflozin improved NO signaling (endothelial eNOS activity, NO availability, cGMP levels, and PKG signaling) and increased titin phosphorylation. Empagliflozin recovered diastolic function, inhibited tissue and molecular remodeling, and inhibited cardiomyocyte stiffness [71].

Although the underlying mechanisms of action of SGLT2 inhibitors in cardio protection are not firmly established, cGMP signaling involvement in such properties of this drug class is plausible as follows; (1) SGLT2 inhibitors promote Na^+^ and glucose excretion (Figure 3), thereby reducing cardiac afterload and ultimately improving NO–cGMP signaling in endothelial cells and stimulating the recovery of endothelial function [72]. (2) SGLT2 inhibitors shift metabolism from glucose to ketone bodies, which bind to and activate G protein-coupled receptors (GPCRs), including GPR43, GPR41, and GPR109A [73]. These GPCRs play important roles in cardiac physiology through the regulation of metabolism, immunity, inflammation, and hormone/neurotransmitter secretion [74]. (3) SGLT2 inhibitors possess antioxidant action. Upon activation of melatonin receptors, cGMP signaling increases the expression of nuclear factor erythroid 2–related factor (Nrf), which exhibits an antioxidant effect [75]. Nrf expression may be induced by the stimulation of ketone receptors through cGMP signaling. (4) SGLT2 inhibitors increase AMP levels and consequently activate 5′ AMP-activated protein kinase (AMPK), which is a suppressor of ischemia-reperfusion injury [76]. Moreover, SGLT2 inhibitors have antiapoptotic effects [77]. However, whether the inhibition of apoptosis is induced in a cGMP signaling-dependent manner by SGLT2 inhibitors is unknown. Nevertheless, the involvement of these cGMP signaling pathways in SGLT2 inhibitor-mediated cardiac protection should be confirmed in future studies of experimental models of chronic HF. In addition, an investigation of combination therapy using an SGLT2 inhibitor and a drug that enhances cGMP signaling during HF will be necessary for demonstrating whether cGMP signaling participates in the cardioprotective function of SGLT2 inhibitors.

HF is often accompanied by titin-dependent cardiomyocyte stiffening. Phosphorylation of titin by PKGI increases myocyte stretching. The activation of GC-B–cGMP–PKGI signaling by CNP in cardiomyocytes was reported to play a protective role in preventing titin-based cardiomyocyte stiffening during the early phase of pressure overload [78]. Empagliflozin attenuated inflammation and oxidative stress, resulting in the recovery of NO–sGC–cGMP signaling in the heart of HFpEF patients. Moreover, the oxidation and polymerization of PKGIα were reduced and cardiomyocyte stiffness was delayed via the recovery of PKGIα activity [79]. In both cases, the mechanism through which SGLT2 inhibition in the kidney evoked protective effects on the heart was not demonstrated. Therefore, the identification of factors that transmit action from the kidney to the heart could advance the development of therapeutic agents for HF.

## 4. β Adrenergic Receptors (βARs) and NO System Signaling

Adrenergic receptors are classified into three subfamilies, α1, α2, and β that couple to Gq, Gi, and Gs proteins, respectively. Each subfamily is further divided into three subtypes. βARs are classified into three subtypes, β_1_, β_2_ and β_3_, all of which mainly couple to the Gs protein. However, β_2_AR and β_3_AR can couple to Gi protein and endothelial-type nitric oxide synthase (eNOS) (possibly via Gi), respectively, depending on cellular conditions [80].

Activation of β_2_AR and β_3_AR is associated with cardiac protection. β_1_ARs regulate cardiac contractility and increase HR via Gs–PKA signaling. Excessive β_1_AR-mediated signaling adversely affects the heart. Persistent and prolonged β_1_AR stimulation triggers HF progression [81]. Sympathetic stimulation is activated in HF because of the reduced cardiac function during HF [81]. Carvedilol not only inhibits excessive catecholamine stimulation but also activates β-arrestin-mediated epidermal growth factor receptor signaling, which protects the heart against stress. β-Arrestin has been identified as a molecule involved in the desensitization of GPCRs [82]. Carvedilol-stimulated cardioprotective signaling via β-arrestin also requires Gi [83]. Later, it was demonstrated that β-arrestin itself acts not only as a cardioprotective molecule but also as a scaffold molecule that mediates the generation of various downstream signals [84,85]. It is reasonable that β-arrestin acts as a scaffold molecule, as it binds almost all GPCRs. However, the currently prescribed drugs acting on β_1_AR are antagonists such as carvedilol and bisoprolol [86]. The effects of βAR antagonists include the inhibition of excessive β_1_AR stimulation. However, β_2_AR stimulation and other receptor-independent actions play a role in the cardioprotective effects of βAR antagonists.

Unlike β_1_ARs and β_2_ARs, β_3_ARs protect the heart against stress-induced hypertrophy and HF by activating eNOS (also called NOS3) and neuronal-type nitric oxide synthase (nNOS, also called NOS1). eNOS and nNOS generate NO, which activates cGMP signaling [87]. nNOS is activated by Ca^2+^-calmodulin and phosphorylated by Ca^2+^-calmodulin-dependent kinase. eNOS is phosphorylated and activated by Akt, whose activity is stimulated by phosphatidylinositol-(3, 4, 5)-trisphosphate (PIP_3_), a product of phosphatidylinositol-3-phosphate kinase (PI3K). Since β_3_AR activates PI3K, β_3_AR stimulation enhances eNOS activity [88]. Mirabegron is a β_3_AR agonist used for the treatment of overactive bladder [89]; however, its application in the treatment of HF was considered. A total of 22 HF patients with New York Heart Association (NYHA) class III were randomly assigned to control and mirabegron groups. The results showed no significant differences between the two groups after administration of mirabegron for a short time. On the other hand, one week of treatment resulted in significant responses in cardiac index and stroke volume [90]. Because of the small number of patients in this trial, further clinical trials with larger numbers of patients are necessary.

In addition to β_3_ARs, β_2_ARs couple with cGMP-producing systems (eNOS and nNOS) and increase cGMP. cGMP increases the phosphorylation of several intracellular proteins via PKG [90,91]. PKG enhances hyperpolarization by activating Na^+^/K^+^-ATPase. It reduces intracellular Ca^2+^ concentrations by suppressing voltage-gated L-type Ca^2+^ channels. It also enhances the Ca^2+^-associated action of phospholamban. The effects of cGMP on PDEs have also been reported. The binding of cGMP to PDE2 stimulates the cAMP-degrading activity of PDE2; the cardioprotective effects of β_2_ARs and β_3_ARs may be mediated by these actions [90,91]. β_2_ARs and β_3_ARs activate eNOS and nNOS by coupling to Gi and enhancing NO–cGMP signaling. However, it has been reported that β_1_AR also increases cGMP production upon stimulation with the antagonist carvedilol [92]. Each βAR subtype increases cGMP in the discrete domain, thus the intracellular compartment of cGMP may be important for the protective effects of βARs.

## 5. Candidate Molecular Therapeutic Targets for HF and Cardiac Remodeling

Several molecules have been reported to have positive effects on cardiac hypertrophy and/or remodeling in vitro or in vivo. In this Section, promising molecules for HF treatment are discussed and summarized, although their relationship to cGMP signaling is unclear.

### 5.1. Free Fatty Acid Receptors (FFARs)

Fatty acids have been shown to exert cardiac effects by acting on GPCRs [93]. GPR40 (known as free fatty acid receptor 1; FFAR1), GPR43 (FFAR2), GPR41 (FFAR3), and GPR120 (FFAR4) are fatty acid receptors. FFAR2 and FFAR3 are activated by short-chain fatty acids such as acetone, butyric acid, and propionic acid, while FFAR1 and FFAR4 are activated by long-chain fatty acids such as α-linolenic acid. These receptors regulate glucose metabolism and inflammation [93].

Eicosapentaenoic acid (EPA) and docosahexaenoic acid (DHA) are omega-3 polyunsaturated fatty acids that play an important role in maintaining CV health [94]. Even though EPA and DHA are agonists of FFAR4, EPA–but not DHA–has been shown to inhibit fibrosis under pressure loading [26,95]. FFAR4 activation inhibits TGF-β mediated fibrosis [26]. Several lines of evidence have demonstrated that TGF-β signaling plays a major role in fibrosis and thus inhibition of TGF-β signaling results in the suppression of fibrosis [96,97]. In fibroblasts, FFAR4 stimulation activates the cGMP–PKG pathway, and thus the phosphorylation of Smad3 by PKG inhibits TGF-β-mediated fibrosis [26]. Signaling from FFAR4 to cGMP production has not been demonstrated. FFAR4 is a Gq-coupled receptor; therefore, a possible mechanism is that Gq-mediated Ca^2+^ elevation activates eNOS and consequently triggers NO production, leading to an increase in cGMP levels. However, activation of Gq in cardiac fibroblasts incompletely unravels the inhibition of TGF-β signaling mediated by cGMP.

A recently developed powerful tool for analyzing specific G protein-mediated actions is the Designer Receptors Exclusively Activated by Designer Drugs (DREADDs) [98]. Transgenic mice expressing Gq-selective DREADD (Gq-DREADD) under the promoter of the muscle creatinine kinase were generated [99]. Treatment of these specific transgenic mice with clozapine-N-oxide, an agonist of all DREADDs, induced arrhythmias. On the other hand, stimulation of Gq-coupled receptors, including α_1_-adrenergic receptor and AT_1_R, expressed in cardiomyocytes prepared from neonatal rat hearts induced hypertrophic responses [100]. In addition, stimulation of AT_1_R of fibroblasts increased expression and extracellular release of TGF-β and connective tissue growth factor in the heart. These released factors directly acted on cardiomyocytes and induced a hypertrophic response [101]. These results indicated that Gq activation does not entirely clarify the cGMP-mediated actions of FFAR4 and further analysis of the intracellular signaling linking FFAR4 to cGMP signaling is still required.

Fatty acid receptors inhibit inflammation in many cells. Since inflammation plays an important role in cardiac remodeling, fatty acid receptors may inhibit remodeling by suppressing inflammatory mediators [102].

### 5.2. Cannabinoids

The role of cannabinoid signaling in human health and disease has been comprehensively examined. Cannabinoid signaling consists of two main receptors, cannabinoid receptor type 1 (CB1) and type 2 (CB2), endogenous ligands, and metabolic enzymes [103]. CB2 ligands play a beneficial immunomodulatory role without inducing CB1-mediated psychotropic effects [103]. Using CB2-knockout mice and CB2-selective ligands, it has been reported that CB2 plays a protective role in cardiovascular diseases. CB2 inhibited adenylyl cyclase via Gi/Go proteins. However, it also activated other important downstream signaling molecules, including MAPK, PI3K, PLC, and Janus kinase and signal transducer and activator of transcription (JAK/STAT) [104]. Treatment with JWH-133, an agonist of CB2, inhibited the infiltration of neutrophils into infarcted areas after ischemia-reperfusion [105]. The CB2 agonist AM1241 improved cardiac function and decreased collagen deposition in addition to decreasing the infarcted area after ischemia-reperfusion [106,107]. Although CB2 signaling is not directly associated with cGMP signaling, the potential benefit of ligands acting on GPCRs in failing hearts warrants further investigation.

### 5.3. Transient Receptor Potential Cation Channel Subfamily V Member 1 (TRPV1) Channel

The transient receptor potential (TRP) channel family is a non-selective cation channel with Ca^2+^ permeability [108]. TRPV channels are characterized by temperature sensitivity; they are activated at high temperatures that cause tissue damage. The main TRP channels that sense temperature alterations include TRPV subtype 1 (TRPV1), TRPV4, transient receptor potential melastatin 3 (TRPM3), TRPM5, TRPM8, transient receptor potential ankyrin 1 (TRPA1), and transient receptor potential canonical 5 (TRPC5). Of these, TRPV1 is strongly associated with fibrosis during cardiac remodeling [108]. However, when GPCRs are activated by multiple inflammatory cascades such as phosphatidylinositol 4, 5–bisphosphate (PIP2) hydrolysis or PKC/PKA phosphorylation, TRPV1 becomes activated at body temperature. Capsaicin and acid stimulation (protons) also activate TRPV1. It is also activated by endogenous cannabinoids, metabolites of the arachidonic acid cascade, and camphor [108,109].

TRPV1 activity is regulated by cGMP signaling. TRPV1 is proposed as a component of the ANP, cGMP, and PKG signaling complex [103]. TRPV1 interacts with NPR-A and its ligand, ANP. When ANP binds to NPR-A, it inhibits TRPV1 activation by producing cGMP and phosphorylating TRPV1 channels via PKG. Furthermore, the administration of TRPV1 inhibitors suppressed ventricular hypertrophy and improved in vivo cardiac function in mice exposed to pressure overload caused by transverse aortic stenosis [109].

### 5.4. Aquaporins

Aquaporins not only work as channels for water molecules but also allow ROS to pass through intracellularly [110]. Aquaporin-1 is expressed in the heart and has been shown to facilitate cellular ROS uptake [110]. Compounds that inhibit the action of aquaporins may be promising due to their ability to block an increase in intracellular ROS. Although aquaporin-2, -4, and -5 are phosphorylated by PKG, phosphorylation of aquaporin-1 by PKG was not observed [111]. Since cGMP signaling regulates aquaporin activity, aquaporins are promising targets of cGMP signaling and would be potential therapeutic targets for HF treatment.

## 6. Clinical Studies of Drugs and Therapeutic Targets for HF Treatment

### 6.1. Clinical Studies of ARNI

LCZ696 (sacubitril/valsartan) is a first-in-class drug that belongs to a group of ARNIs, which is a combination of two drugs, valsartan, and sacubitril, in a fixed dose. LCZ696 targets the dual renin-angiotensin system and natriuretic peptide metabolism to treat hypertension and HF [40]. Valsartan is a well-established ARB that inhibits the action of Ang II by blocking AT_1_R, leading to a reduction in vasoconstriction and aldosterone production, providing CV benefits [41]. Sacubitril (AHU377) is a neprilysin inhibitor that inhibits neprilysin enzymatic activity, which in turn prevents the degradation of natriuretic peptides such as ANP, BNP, and CNP, leading to augmentation of these peptides that possess blood-lowering properties [41]. LCZ696 is the only drug in the ARNI group that is currently available and approved (since 2015) by the United States Food and Drug Administration (US FDA) and has shown benefits in patients with chronic HF.

The PARAMOUNT study (Table 6) was a phase II, randomized, double-blind, parallel-group trial conducted in 301 HFpEF patients who had HF with LVEF ≥ 45%, NYHA class II–III symptoms, and NT-proBNP > 400 pg/mL [112]. These patients were assigned to receive either LCZ696 (200 mg twice daily; BID) or valsartan (160 mg BID). Analysis of the primary endpoint was carried out by monitoring changes in NT-proBNP at baseline and after 12 weeks of treatment. After 12 weeks of therapy, NT-proBNP levels were significantly reduced in patients receiving LCZ696 (baseline versus 12 weeks; 783 vs. 605 pg/mL) compared with those receiving valsartan (baseline vs. 12 weeks; 862 vs. 835 pg/mL) (HR 0.77; 95% CI: 0.64–0.92; *p* = 0.005). This reduction in NT-proBNP levels was sustained from 12 to 36 weeks of treatment. Additionally, HFpEF patients receiving LCZ696 for 36 weeks also showed a decrease in left atrial size and an improvement in HF symptoms [112].

The efficacy of LCZ696 on a composite CV death or a first hospitalization with HF (primary endpoint) was also compared with enalapril, an ACE inhibitor, in a larger population of 8399 patients with HFrEF monitored in the PARADIGN-HF trial [113]. In this approach, all patients with LVEF ≤ 40%, NYHA class II–IV, plasma BNP ≥ 150 pg/mL, and NT-proBNP level ≥ 600 pg/mL were randomized to receive LCZ696 (200 mg BID) or enalapril (10 mg BID). The data demonstrated that LCZ696 had greater effects in reducing the incidence of CV death compared with enalapril (HR 0.80; 95% CI 0.71–0.89; *p* < 0.001). However, there was a higher occurrence of angioedema and hypotension in patients treated with LCZ696 compared with those receiving enalapril (14% versus 9.2%) [113].

The impact of two drugs, sacubitril and valsartan, on the incidence of a composite outcome of total HF hospitalizations and deaths from CV causes was further investigated in the PARAGON-HF trial, a phase III, double-blind trial [114]. A total of 4822 HFpEF patients with LVEF ≥ 45%, NYHA class II–III, and elevated levels of NT-proBNP were treated with either LCZ696 or valsartan. The primary outcome of the study indicated that there was no significant difference in the primary event (HR 0.87; 95% CI: 0.75–1.01; *p* = 0.06), especially the rates of HF hospitalizations and CV death. However, the LCZ696 group had a lower incidence of hyperkalemia and higher incidences of angioedema and hypotension compared with the valsartan group. These findings raise concerns regarding the potential side effects of LCZ696 in HFpEF patients [114].

In the 2022 AHA/ACC/HFSA guideline for the management of HF [47], ARNI is shown to reduce morbidity and mortality in patients with HFrEF (Class 1 recommendation). In addition, for patients with HFrEF who tolerate ACEIs or ARBs, replacement with ARNI is recommended to further reduce morbidity and mortality (Class 2b recommendation). Furthermore, ARNI may be considered to reduce hospitalization of patients with HFpEF [1].

### 6.2. Clinical Studies of HCN Channel Blocker

A high HR can increase various CV risks and disrupt the balance of oxygen supply to the heart, leading to heart ischemic conditions and CV death [115]. Ivabradine is a HCN channel blocker that selectively binds to the *I*_f_ channel, inhibiting the *I*_f_ current (also known as the pacemaker current or funny current), which plays a crucial role in controlling HR. Specifically, ivabradine targets HCN4, the primary isoform found in the SA node of the heart, by entering the channel and blocking it from inside [53,54]. Inhibiting *I*_f_ with ivabradine results in decreased SA node depolarization, leading to a lower HR and improved myocardial perfusion without altering myocardial contractility and electrophysiological properties, thus providing the pure HR lowering effect [116,117,118]. Ivabradine is a specific US FDA-approved HR-reducing agent that has been used in humans since 2015. It is well-tolerated and effective in the treatment of coronary artery disease (CAD) and HF [115]. To date, the BEAUTIFUL and SHIFT trials (Table 7) have provided evidence supporting the potential benefits of ivabradine when added to the background therapy for HF patients.

The BEAUTIFUL trial was a double-blind, placebo-controlled trial that sought to examine the HR-lowering effect of ivabradine on CV death and morbidity [117]. The study included 10,917 patients with stable CAD and LVEF < 40% who received either ivabradine at a dosage of 5–7.5 mg or placebo BID. The results demonstrated no significant differences in a composite of CV death and admission to hospital for acute myocardial infarction or HF (primary endpoint) between the ivabradine group (15.4%) and placebo (15.3%) group (HR 1.00; 95% CI 0.91–1.10; *p* = 0.94) [117]. However, the study did identify a subgroup of patients with HR ≥ 70 bpm, in which ivabradine demonstrated advantages in improving outcomes associated with CAD by lowering coronary revascularization by 30% (HR 0.70; 95% CI 0.52–0.93; *p* = 0.016). The BEAUTIFUL echo sub-study aimed to examine the impact of ivabradine on the LV end-systolic volume index (LVESVI), which is a parameter used to assess the size and function of LV, and specifically focused on patients with stable CAD and LV systolic dysfunction [118]. A total of 590 patients were randomized to receive ivabradine (5–7.7 mg BID) or placebo, and the primary endpoint was compared with LVESVI at baseline between 3 and 12 months. The results showed that treatment with ivabradine led to a decrease in LVESVI (−1.48 ± 13.00 mL/m^2^), while the placebo group showed an increase (1.85 ± 10.54 mL/m^2^) (*p* = 0.018). Notably, this reduction was found to depend on the degree of HR reduction achieved in patients. Additionally, ivabradine had a greater ability to increase LVEF (2.00 ± 7.02%) compared with the placebo (0.01 ± 6.20%) (*p* = 0.009). These findings indicated the beneficial effects of ivabradine on LV remodeling in patients with CAD and LV systolic dysfunction [118].

A captivating study involving 6558 HF patients with moderate-to-severe symptoms and LV systolic dysfunction was undertaken in the SHIFT trial [119]. The trial initiated patient recruitment specifically targeting those with symptomatic HF and LVEF ≤ 35% combined with HR ≥ 70 bmp and examined the effect of ivabradine on the composite of CV death or hospital admission for worsening HF (primary endpoint). The results showed that, compared with the placebo, ivabradine led to a reduction in HR, with a net reduction of 9.1 bmp after 1 year of follow-up. Importantly, patients who received ivabradine exhibited a significant reduction in the primary endpoint event compared with those on placebo (HR 0.82; 95% CI: 0.75–0.90; *p* < 0.0001) [119]. However, it is important to note that ivabradine did not demonstrate the same level of efficacy in reducing CV deaths and all-cause deaths [113].

The results of the SHIFT echocardiography sub-study proved the beneficial outcomes of ivabradine on the reversal of cardiac remodeling (Table 7). This sub-study involved 411 chronic HF patients with LVEF ≤ 35% and HR ≥ 70 bpm and studied the effect of ivabradine on LV end-systolic volume index (LVESVI) over an 8-month period and compared the results with the baseline (primary sub-study endpoint) [120]. The findings revealed that ivabradine was superior to placebo, reducing LVESVI by 15% or more (difference (SE), −5.8; 95% CI: −8.8 to −2.7; *p* < 0.001), indicating a decrease in LV volume. Additionally, treatment with ivabradine demonstrated improvements in CV parameters. Specifically, it increased LVEF by 2.4% ± 7.7% (*p* < 0.001) and resulted in a decrease in LV end-diastolic volume index (LVEDVI) from a baseline value of 93.9 ± 32.8 mL/m^2^ to a lower value of 85.9 ± 30.9 mL/m^2^ (*p* = 0.002) at the 8-month mark. These suggested that ivabradine has the benefit of LV remodeling patients with HF and LV systolic dysfunction [120]. Following the results of these clinical studies, ivabradine was approved by the US FDA and indicated (1) to reduce the risk of hospitalization with worsening HF in adults and (2) for the treatment of stable symptomatic HF caused by dilated cardiomyopathy in patients aged 6 months and older.

### 6.3. Clinical Studies of Cardiac Myosin Activators

Cardiac contraction relies on the function and cross-bridging cycle of myosin and actin filaments in cardiomyocytes, and the impairment of cardiac contractility is a significant problem in systolic HF progression [121]. Cardiac myosin activators are a new class of drugs that have the ability to increase heart contractility and improve cardiac performance, minimizing the adverse effects commonly associated with older inotropic agents [57]. These drugs directly bind to and activate the cardiac isoform of myosin, promoting cardiac myosin ATP hydrolysis without altering calcium homeostasis during the cardiac cycle. This, in turn, enhances myosin and actin interaction, leading to increased myocyte contraction and thus increasing cardiac output [57,58]. Omecamtiv mecarbil and danicamtiv are novel cardiac-selective myosin activators currently under clinical trial investigation for their potential role when used in combination with standard therapy for HF treatment [122,123].

To date, several clinical studies have demonstrated the beneficial effects of omecamtiv in HFrEF patients (Table 8). The phase II COSMIC-HF trial aimed to investigate the pharmacokinetic (PK) effects of omecamtiv on cardiac function and structure in HFrEF patients [124]. The trial involved administering omecamtiv at a fixed dose (25 mg BID) and PK titration (25 mg BID titrated to 50 mg BID) over a period of 20 weeks. The results showed that the mean maximum concentration of omecamtiv at 12 weeks was higher in the PK-titration group (318 ng/mL) compared with the fixed-dose group (200 ng/mL). Moreover, the PK-titration group demonstrated significant improvements in cardiac function by decreasing LVES diameter (*p* = 0.0027), LVED diameter (*p* = 0.0128), NT-proBNP level (*p* = 0.0069), and HR (*p* = 0.007) compared with the placebo. These findings suggested that omecamtiv dosing guided by PK enhanced cardiac performance and promoted favorable ventricular remodeling in HFrEF patients [124].

The COSMIC-HF study provided additional data on the effect of omecamtiv on HF symptoms and health-related quality of life (HRQoL) in HFrEF patients [125]. The effect of omecamtiv on HF symptoms was more pronounced in patients with severe HF symptoms compared with those with mild symptoms over a period of 20 weeks. Furthermore, the omecamtiv-PK titration group exhibited improved HRQoL, as indicated by higher TSS scores of Kansas City Cardiomyopathy Questionnaire (KCCQ), in comparison to the placebo group (*p* = 0.03), with an association with a decrease in NT-proBNP levels. These insights highlight the beneficial effects of omecamtiv treatment on the overall well-being and quality of life of HF patients [125].

Measuring exercise capacity is crucial in the management of HF as it allows for the assessment of the severity of the condition and HF progression. Improving exercise capacity is a key goal for HF patients who commonly experience exercise intolerance [126]. In this context, a phase III METEORIC-HF trial was conducted to investigate the impact of omecamtiv on changes in exercise capacity (primary endpoint; maximum oxygen consumption; peak Vo_2_) in HFrEF patients [127]. The results revealed that there was no significant difference in the change in peak Vo_2_ between the group receiving omecamtiv (−0.24 mL/kg/min) and the placebo group (0.21 mL/kg/min) from baseline to a period of 20 weeks. Similarly, no other secondary endpoints were observed, suggesting that omecamtiv did not improve exercise capacity in HFrEF patients [127].

**Table 8 ijms-24-12866-t008:** Clinical studies of cardiac myosin activator.

Drug	Study Population	Treatment	Primary and Secondary Endpoints	Main Findings and Conclusions
Danicamtiv [123]	▪ HFrEF patients with LV-EF on echocardiography of 40% or lower (N = 40)	Danicamtiv 50, 75, or 100 mg BID or placebo for 7 days	Primary: ▪ Safety and tolerability of single and multiple doses. Secondary: ▪ Stroke volume, fractional shortening, ejecting time.	▪ Danicamtiv increased stroke volume and LA function index. ▪ Danicamtiv improved global longitudinal and circumferential strain. ▪ Danicamtiv improved LV volume and function.
Omecamtiv (COSMIC-HF trial) [124,125]	▪ HFrEF patients with LVEF ≤ 40% ▪ NT-proBNP at least 200 pg/mL (N = 448)	Omecamtiv 25 mg BID (fixed-dose), 25 mg BID titrated to 50 mg BID or placebo for 20 weeks	Primary: ▪ Changes in cardiac function and ventricular diameter.	▪ Omecamtiv improved cardiac function and decreased ventricular diameter. ▪ Reduction in HR and NT-proBNP levels with omecamtiv.
			▪ Evaluate the effects of omecamtiv on symptoms and HRQoL.	▪ Omecamtiv improved HRQoL in HFrEF patients assigned to the pharmacokinetic-titration group.
Omecamtiv (METEORIC-HF trial) [127]	▪ HFrEF patients with LVEF < 35% ▪ NYHA II–III ▪ NT-proBNP level > 200 pg/mL ▪ Peak oxygen uptake (Vo_2_) < 75% (N = 276)	Omecamtiv 25, 37.5, or 50 mg BID or placebo for 20 weeks	Primary: ▪ Change in exercise capacity (peak Vo_2_). Secondary: ▪ Total workload. ▪ Ventilatory efficiency and daily physical activity	▪ No significant difference in the improvement of exercise capacity over 20 weeks.
Omecamtiv (GALACTIC-HF trial) [128,129,130]	▪ HFrEF patients with LVEF ≤ 35% ▪ NYHA II–IV ▪ NT-proBNP level ≥ 400 pg/mL (N = 8256)	Omecamtiv 25, 37.5, or 50 mg BID based on target plasma level or placebo for 20 weeks	Primary: ▪ A composite of a first HF event or death from CV causes. Secondary: ▪ CV death. ▪ Change in TSS ▪ First HF hospitalization.	▪ No significant difference in the change in TSS between groups. ▪ At week 24, the change in NT-proBNP level was 10% lower in the omecamtiv group. ▪ Omecamtiv had a lower occurrence of a composite of HF event or death from CV causes.
			▪ Evaluate the effect of omecamtiv on baseline EF.	▪ Omecamtiv had a greater reduction in HF events in patients who have lower EF at baseline. ▪ Omecamtiv produced greater therapeutic benefit as baseline EF decreased.
			▪ Evaluate the effect of omecamtiv on NT-proBNP level.	▪ Omecamtiv had a greater effect on the primary outcome in patients who had higher NT-proBNP levels at baseline.

GALACTIC-HF was a phase III, double-blind, placebo-controlled trial conducted to investigate the effectiveness and safety of omecamtiv in HFrEF patients [128]. Within this framework, a total of 8256 patients with symptoms of HF, LVEF ≤ 35%, and NYHA class II–IV were enrolled to investigate the effect of omecamtiv on a composite of HF event or CV death (primary endpoint). Compared with the placebo group (39.1%), the omecamtiv with PK-guided group (37.0%) (HR 0.92; 95% CI 0.86–0.99; *p* = 0.03) demonstrated a significant decrease in the incidence of the primary outcome without increasing the risk of clinical adverse effects. However, the use of omecamtiv did not lead to improvements in secondary outcomes such as changes in the KCCQ score and various CV death [128]. In 2021, the same group of investigators [129] conducted the GALACTIC-HF study to investigate the impact of baseline EF on the therapeutic effect of omecamtiv in HFrEF patients. The primary composite outcome was a composite of the occurrence of CV death and HF event. The study revealed that omecamtiv had greater treatment effects as baseline EF decreased. Patients with a baseline EF ≤ 22% had a 17% relative risk reduction of the primary outcome compared with those with EF ≥ 33% (*p* = 0.004). This suggested that omecamtiv had stronger therapeutic benefits in HFrEF patients with lower baseline EF levels [129].

Recently, another study utilized the GALACTIC-HF trial to evaluate the efficacy of omecamtiv in HFrEF patients based on their baseline NT-proBNP levels, which were divided at the median (≤median, >median) [130]. Omecamtiv improved a composite of worsening HF events or CV death (primary outcome) in patients with NT-proBNP level > median compared with patients with NT-proBNP level ≤ median. Importantly, this beneficial effect was more pronounced in patients without atrial fibrillation/flutter compared with the overall population [130]. Furthermore, treatment with omecamtiv resulted in a significant reduction in NT-proBNP levels from baseline after 24 and 48 weeks, specifically in patients with NT-proBNP levels > median (HR 0.81; 95% CI: 0.73–0.90; *p*-interaction = 0.095). These findings highlight that the most prominent benefits of omecamtiv were observed in HFrEF patients with high levels of NT-proBNP [130]. However, the US FDA has declined to approve omecamtiv mecarbil for the treatment of HFrEF patients. Additional clinical trials of omecamtiv are required to establish evidence of effectiveness in HFrEF patients.

In addition to omecamtiv, danicamtiv has been reported to have beneficial effects in pre-clinical as well as clinical studies involving HFrEF patients (Table 8). In dogs with HF, danicamtiv increased ATPase activity and Ca^2+^ sensitivity in muscle fibers and myofibrils taken from the left atrial (LA) and LV chambers. It also improved LV stroke volume and LA emptying fraction in dogs with HF [123]. A phase IIa, double-blind clinical trial evaluated the effects of danicamtiv on HFrEF patients and showed that danicamtiv (at plasma concentrations ≥ 2000 ng/mL) improved stroke volume (*p* < 0.01), myocardial strain (*p* < 0.01), and LA function index (*p* < 0.01), while decreased LA minimal volume index (*p* < 0.01), suggesting enhanced cardiac output and atrial function [123].

### 6.4. Clinical Studies of SGLT2 Inhibitors

SGLT2 is the predominant transporter that is responsible for glucose reabsorption from the proximal renal tubule back into blood circulation. Blocking SGLT2 activity resulted in a reduction of glucose reabsorption and thus increased urinary glucose excretion. For this major reason, SGLT2 inhibitors (e.g., canagliflozin, dapagliflozin, and empagliflozin) are approved for the treatment of type 2 diabetes mellitus (T2DM) [131]. In addition to antihyperglycemic effects, several cardioprotective properties of SGLT2 inhibitors have been demonstrated in preclinical and clinical studies in patients with HFrEF or HFpEF. Such cardioprotective effects include improved myocardial functions and energy metabolism, reduced afterload and preload; lowered BP, natriuresis, and diuresis; and reduced cardiac hypertrophy, fibrosis, and remodeling [131,132].

The recent canagliflozin CHIEF-HF clinical trial was conducted in HF patients regardless of EF or diabetes status [133] (Table 9). A total of 476 HF patients were randomized to receive a placebo or 100 mg of canagliflozin once daily. After 12 weeks, canagliflozin improved KCCQ TSS compared with the placebo, meeting the primary endpoint. In addition, canagliflozin was associated with the improvement of symptoms and quality of life within 12 weeks in HF patients with either reduced or preserved EF and in those with or without diabetes [133]. Thus, the CHIEF-HF trial demonstrated the benefits of canagliflozin in improving patients’ symptoms regardless of EF or diabetes status.

In addition to reducing the risk of hospitalization with HF in patients with T2DM, dapagliflozin is indicated by the US FDA to reduce the risk of CV deaths and hospitalization in HFrEF patients with NYHA class II–IV. The expansion of the indication of dapagliflozin is based on clinical data from many trials conducted in HF patients (e.g., DELIVER, DAPA-HF, and DEFINE-HF). The DAPA-HF trial [134] was a double-blind, placebo-controlled trial investigating the efficacy and safety of dapagliflozin in HFrEF patients with or without diabetes (Table 9). The primary outcome was a composite of worsening HF or CV death and the results showed that 10 mg/day of dapagliflozin was associated with a greater reduction in the risk of worsening HF or death from CV causes and had better symptom scores compared with the placebo group, regardless of diabetic status [134]. Data from the DAPA-HF study concluded that dapagliflozin was effective at reducing the incidence of the primary endpoint and was well-tolerated in patients with HFrEF regardless of diabetic status [134].

The DEFINE-HF trial was a double-blind, placebo-controlled trial conducted in HFrEF patients with LVEF ≤ 40% and NYHA class II–III to investigate the effects of dapagliflozin on symptoms, functional status, and biomarkers [135]. Patients (N *=* 263) were randomized to receive a placebo or 10 mg/day of dapagliflozin for 12 weeks. Dual primary endpoints consisted of a composite of the proportion of patients who achieved improved health status and the average NT-proBNP levels at 6 and 12 weeks. In this trial, there were no significant differences in average adjusted NT-proBNP levels between dapagliflozin and placebo (1133 and 1191 pg/dL, respectively) [135]. However, the dapagliflozin group had a higher proportion of patients with clinically meaningful improvements in HF-related health status. These results were consistent in HFrEF patients with or without T2DM.

Furthermore, the DELIVER trial was conducted in HFpEF patients with or without T2DM to evaluate the efficacy and safety of dapagliflozin [51]. In addition to background therapy, patients were randomized to receive a placebo or 10 mg of dapagliflozin once daily (OD). The primary outcome was a composite of worsening HF or CV death, and the results demonstrated a significant reduction in HF hospitalizations with dapagliflozin and no significant differences in CV deaths compared with the placebo [51]. These data support the use of SGLT2 inhibitors as a therapy in HF patients, regardless of LVEF status or the presence or absence of diabetes.

Empagliflozin is one of the SGLT2 inhibitors indicated by the US FDA to reduce the risk of CV death in adults with T2DM and established CVDs as well as to reduce the risk of CV death and hospitalization with HF. The expansion of the indication for empagliflozin is based on clinical data from many trials in HF and diabetic patients, including CANVAS, DECLARE-TIMI58, EMPA-REG OUTCOME, EMPEROR-Reduced, and EMPEROR-Preserved studies. The EMPEROR-Reduced trial was conducted in HFrEF patients with LVEF ≤ 40%, NYHA class II–IV, and elevated NT-proBNP levels to investigate the efficacy and safety of empagliflozin as an adjunct to background therapy for HF [136] (Table 9). The patients (N *=* 3730) were randomized to receive empagliflozin (10 mg OD) or placebo, and the primary outcome was a composite of CV death or hospitalization with worsening HF. Over 6 months of treatment, a primary outcome event occurred in 19.4% of the empagliflozin group and 24.7% of the placebo group (HR 0.75; 95% CI: 0.58–0.85; *p* < 0.001) [136]. Subgroup analysis demonstrated that the efficacy of empagliflozin was consistent in HFrEF patients in the presence or absence of type 2 diabetes, indicating that empagliflozin had a lower risk of CV death or hospitalization, regardless of diabetic status. In addition, empagliflozin had a slower rate of decline in eGFR than placebo. However, empagliflozin had a higher incidence of uncomplicated genital tract infections than the placebo [136].

Empagliflozin was also investigated for its efficacy and safety in HFpEF patients with LVEF > 40%, NYHA class II–IV, and NT-proBNP levels > 300 pg/mL in the EMPEROR-Preserved study [137]. The 5988 patients were randomized to receive empagliflozin (10 mg OD) or placebo and the primary outcome was a composite of CV death or hospitalization with worsening HF. Over a median of 26.2 months, the primary outcome event was achieved in 13.8% and 17.1% of patients in the empagliflozin and placebo groups, respectively (HR 0.79; 95% CI: 0.69–0.90; *p* < 0.001) [137]. For safety, the incidences of adverse events, including hypotension and uncomplicated genital and urinary tract infections, were higher in the empagliflozin group compared with the placebo group [137]. This EMPEROR-Preserved trial supported the indication of empagliflozin for patients with HFpEF, regardless of diabetic status.

In the 2022 AHA/ACC/HFSA guideline for the management of HF [1], SGLT2 inhibitors are indicated to reduce morbidity and mortality in patients with HFrEF (Class 1 recommendation). In addition, SGLT2 inhibitors could be beneficial for reducing HF hospitalizations and mortality in HFpEF patients (Class 2a recommendation).

### 6.5. Clinical Studies of Soluble Guanylyl Cyclase (sGC) Stimulators/Activators

sGC-mediated cGMP signaling plays a crucial role in regulating normal cardiovascular and cardiopulmonary functions [62]. Activation of sGC by its endogenous ligand, nitric oxide (NO), leads to the production of cGMP, which in turn stimulates various downstream signaling pathways involved in regulating vascular tone, including vasodilation and several cellular functions [62,63]. However, the endothelial dysfunction commonly observed in HF can result in impaired NO production, leading to reduced cGMP production and progression of HF and CVDs [64]. To address this cGMP deficiency, sGC stimulators are used to directly activate sGC and increase intracellular cGMP levels. Various sGC stimulators, including vericiguat, praliciguat, riociguat, and cinaciguat, share the general concept of sGC stimulation and elevation of cGMP levels. However, they differ in their specific binding properties and modes of action, which may have implications for their clinical use and effectiveness in specific patient populations [62]. In 2021, vericiguat was approved by the US FDA and recommended for reducing the risk of CV death and hospitalization in adults with LVEF ≤ 45%. Other sGC stimulators such as praliciguat and riociguat are currently under investigation for their efficacy and safety in HF patients, as shown in Table 10.

The SOCRATES-REDUCED trial was a double-blind, placebo-controlled, phase II trial aimed at studying the effect of vericiguat on NT-proBNP levels in HFrEF patients [138]. Different doses of vericiguat were administered to a total of 456 patients and changes in log-transformed NT-proBNP levels were monitored from baseline to 12 weeks. The results demonstrated that vericiguat (at doses of 2.5 mg, 5 mg, and 10 mg) did not significantly cause any changes in NT-proBNP levels compared with the placebo [138]. However, the exploratory secondary analysis indicated a potential dose–response relationship, indicating that higher doses of vericiguat were associated with greater reductions in NT-proBNP levels (*p* < 0.02) [138].

Similarly, the SOCRATES-PRESERVED trial compared the effects of vericiguat (fixed doses vs. titrated doses) and placebo on changes in NT-proBNP levels and left atrial volume (LAV) over 12 weeks in 447 HFpEF patients [139]. The study showed no significant differences in NT-proBNP levels and LAV between vericiguat and placebo groups. However, vericiguat, particularly at a target dose of 10 mg, showed improved KCCQ scores, indicating an enhancement in the quality of life of patients [139]. In the VITALITY-HFpEF trial, another randomized phase II trial, the efficacy of vericiguat was analyzed by examining changes in the physical limitation score (PLS) of KCCQ in a cohort of 789 HFpEF patients after 24 weeks of treatment [68]. The baseline and mean 24-week KCCQ PLS (primary endpoint) were recorded for three groups: vericiguat (10 or 15 mg/day) and placebo. The results of the study showed that, compared with the placebo, vericiguat did not lead to any improvement in the KCCQ PLS. This suggested that vericiguat treatment did not help HFpEF patients enhance their ability to engage in physical activities and tasks [68].

Results of larger phase III clinical trials showing the efficacy of vericiguat in HFrEF patients were published in 2020 [66,140]. The VICTORIA trial was a randomized, double-blind, placebo-controlled trial involving 5050 HFrEF patients who were randomly assigned to receive either a placebo or vericiguat at a target dose of 10 mg OD [66]. The results showed that, at a median follow-up of 10.8 months, patients who received vericiguat had a significantly decreased incidence of a composite of CV death or HF hospitalization (35.5%) (primary outcome) compared with the placebo (38.5%) (HR 0.90; 95% CI: 0.82–0.98; *p* = 0.02) [66]. Furthermore, the same group of investigators also studied the effect of vericiguat in relation to NT-proBNP levels and the primary outcome of CV death or HF hospitalization in the VICTORIA trial [140]. The study revealed a significant interaction between treatment effects and NT-proBNP levels, particularly in patients with NT-proBNP levels up to 8000 pg/mL. Notably, there was a 23% reduction in the primary endpoint in patients with NT-proBNP levels ≤ 4000 pg/mL (HR 0.77; 95% CI: 0.68–0.88) [140]. However, no significant difference was observed in patients with NT-proBNP levels > 8000 pg/mL. This finding indicates that the efficacy of vericiguat treatment depends on the levels of NT-proBNP in HF patients, providing additional data to support its use in the HF patient population [140].

In addition to vericiguat, clinical studies have also investigated the efficacy of other drugs belonging to the sGC stimulator group, such as praliciguat [69] and riociguat [141]. The CAPACITY-HFpEF trial was a double-blind, placebo-controlled, phase II trial involving 196 patients with LVEF > 40%, impaired maximum oxygen consumption (peak Vo_2_), and at least two conditions associated with NO deficiency [69]. Patients received praliciguat (40 mg daily) over a period of 12 weeks; the efficacy of praliciguat was then evaluated by assessing changes in peak Vo_2_ (primary endpoint) from baseline. The results showed that treatment with praliciguat (0.04 mL/kg/min) had no significant effect on the primary endpoint compared with the placebo (−0.26 mL/kg/min) [69]. These findings suggest that praliciguat is not effective at treating HFpEF patients.

HF and pulmonary hypertension, a condition characterized by increased BP in the pulmonary arteries, often coexist and can mutually influence each other, impacting heart-related diseases. The LEPHT trial was a phase II trial that aimed to evaluate the hemodynamic effects of riociguat in patients with pulmonary hypertension caused by systolic LV dysfunction [141]. This randomized, double-blind, placebo-controlled trial enrolled 201 patients with LVEF ≤ 40% and mean pulmonary artery pressure (mPAP) ≥ 25 mmHg. The patients were treated with riociguat (0.5, 1, or 2 mg TID) or placebo for a duration of 16 weeks. The results revealed that the highest dose of riociguat showed no significant changes in mPAP (primary endpoint) from baseline to week 16 compared with the placebo [141]. However, riociguat showed significant improvements in cardiac index (*p* < 0.0001), stroke volume index (*p* < 0.0018), systemic vascular resistance (SVR) (*p* = 0.0002), and pulmonary vascular resistance (PVR) (*p* = 0.03) but no effect on HR and BP [141].

The 2022 AHA/ACC/HFSA guideline for the management of HF [1] indicates that the oral sGC stimulator vericiguat can be beneficial in reducing HF hospitalization and CV deaths in high-risk patients with HFrEF and improving recent worsening of HF patients already on guideline-directed medical therapy (Class 2b recommendation). Based on data from animal studies, vericiguat may cause harm to fetuses; thus, vericiguat should not be administered to women with HF who are pregnant or planning to become pregnant [1].

**Table 10 ijms-24-12866-t010:** Clinical studies of sGC stimulator (activator).

Drug	Study Population	Treatment	Primary and Secondary Endpoints	Main Findings and Conclusions
Praliciguat (CAPACITY-HFpEF) [69]	▪ HFrEF patients with LVEF > 40%, impaired peak Vo_2_, and at least 2 conditions associated with NO deficiency (N *=* 196).	Praliciguat 40 mg OD or placebo for 12 weeks	Primary: ▪ Change in peak Vo_2_. Secondary: ▪ Change in 6-min walk test distance and ventilatory efficiency.	▪ No significant difference in the change in peak Vo_2_ from baseline to week 12.
Riociguat (LEPHT) [141]	▪ Patients with HF resulting from pulmonary hypertension. ▪ LVEF ≤ 40%. ▪ mPAP ≥ 25 mm Hg at rest (N = 201).	Riociguat 0.1, 1, or 2 mg TID or placebo for 16 weeks	Primary: ▪ Change in mPAP. Secondary: ▪ Change in hemodynamic and echocardiography parameters.	▪ Primary endpoint (change in mPAP) was not met. ▪ Riociguat improved cardiac index, PVR, SVR, and health-related QoL without altering HR and BP.
Vericiguat (SOCRATES-REDUCED trial) [138]	▪ Patients with LVEF < 40% and a recent episode of worsening chronic HF (N = 456).	Vericiguat 1.25, 2.5, 5, or 10 mg OD or placebo for 12 weeks	Primary: ▪ Change in NT-proBNP levels from baseline to week 12.	▪ No significant difference in the change in NT-proBNP levels between groups. ▪ Higher vericiguat had a greater reduction of NT-proBNP levels.
Vericiguat (SOCRATES-PRESERVED trial) [139]	▪ Patients with symptomatic worsening chronic HF and LVEF ≥ 45% (N = 477).	Vericiguat 1.25–10 mg OD or placebo for 12 weeks	Primary: ▪ Change in NT-proBNP levels and left atrial volume (LAV) from baseline to week 12.	▪ No significant differences in the changes in NT-proBNP levels and LAV. ▪ Vericiguat was associated with improved QoL.
Vericiguat (VITALITY-HFpEF trial) [68]	▪ Patients with chronic HFpEF and LVEF ≥ 45% with NYHA II–III, within 6 months of a recent decompensation (N = 789).	Vericiguat up-titrated to 10 or 15 mg OD or placebo for 24 weeks	Primary: ▪ Change in KCCQ PLS (range 0–100) at 24 weeks. Secondary: ▪ 6-min walking distance.	▪ Vericiguat did not improve the physical limitation score of the KCCQ.
Vericiguat (VICTORIA trial) [66,140]	▪ Patients with chronic HF and LVEF < 45%. ▪ NYHA II–IV. ▪ Elevated natriuretic peptide levels (N = 5050).	Vericiguat 10 mg OD or placebo	Primary: ▪ Composite of death from CV causes or first hospitalization with HF	▪ Among patients with high-risk HF, vericiguat reduced the risk of death from CV causes or hospitalization with HF.
			▪ Evaluate NT-proBNP relationship with the primary outcome	▪ Vericiguat showed a reduction in CV deaths or HF hospitalization in patients with NT-proBNP levels up to 8000 pg/mL.

### 6.6. Clinical Studies of β_3_ Adrenergic Receptor (β_3_AR) Agonists

Mirabegron is an agonist of β_3_AR. Stimulation of β_3_ARs leads to the relaxation of the detrusor smooth muscle of the urinary bladder during the storage phase, resulting in increasing bladder capacity [142]. Due to this effect, mirabegron was approved for the treatment of overactive bladders. Interestingly, β_3_ARs are expressed in the heart, and stimulation of β_3_ARs exhibits cardioprotective effects, including improved systolic functions.

The BEAT-HF trial was the first clinical study of mirabegron in HFrEF patients with LVEF < 40% and NYHA class II–III (Table 11). A total of 70 patients were randomized to receive placebo or mirabegron titrated to 150 mg twice daily for 6 months and the LVEF change from baseline to 6 months was used as a primary endpoint [143]. There was no significant difference in changes in LVEF after 6 months on mirabegron versus placebo (*p* = 0.82) [143]. However, in an exploratory analysis, mirabegron increased the mean LVEF of patients with more severe HF at baseline (*p* < 0.001) (baseline LVEF < 40% as measured using computed tomography). Based on the safety profile, mirabegron was generally well tolerated. Due to the small sample size of the BEAT-HF trial, additional studies on the effect of mirabegron in patients with HFrEF are needed.

The most recent trial of mirabegron efficacy and safety was the BEAT-HF-II trial, which was conducted in HFrEF patients with LVEF < 35% and NYHA class III–IV and measured the hemodynamic response to mirabegron [90]. Patients were randomized to receive mirabegron (300 mg/day) or placebo for a week; the invasive hemodynamic parameters were changes in stroke volume, cardiac index, BP, HR, pulmonary vascular resistance (PVR), and systemic vascular resistance (SVR) (Table 11). After one week, mirabegron treatment was associated with significant improvements in cardiac index (mean difference, 0.41; 95% CI: 0.07–0.75; *p* = 0.039) and PVR (mean difference, −1.6; 95% CI: −0.4 to −2.8; *p* = 0.02) [90]. There were no differences in changes in BP, HR, and SVR between the mirabegron and placebo groups [90]. In conclusion, based on the favorable safety, the increase in cardiac index, and the decrease in peripheral vascular resistance, the β_3_AR-agonist mirabegron is one of the therapeutic targets for the treatment and prevention of HF. However, mirabegron needs to be investigated in phase II clinical studies conducted over a longer duration and with larger numbers of HF patients.

## 7. Conclusions and Further Directions

cGMP signaling plays a crucial role in various pathophysiological processes, including HF and remodeling; thus, the identification of new molecules targeting cGMP signaling lead to the development of additional treatments for HF. The phosphoproteomic approach, a branch of proteomics that focuses on the identification and quantification of the phosphorylation process of proteins, in particular biological samples, is a valuable tool for searching for new targets in cGMP signaling [144]. Since no signaling molecule corresponding to Epac of cAMP has been found, it is reasonable to assume that cGMP exerts its effects mainly via PKG. Therefore, phosphoproteomics can be used to comprehensively search for proteins whose phosphorylation is upregulated by activation of cGMP–PKG signaling. Consequently, target molecules capable of inhibiting the cGMP signaling-dependent cardiac hypertrophic response and remodeling are proposed as promising drug candidates for HF. Over the past few years, there have been significant advancements in the treatment of HF, with the development of several novel drug classes showing promising results in both animal and clinical trials. Some of the newly developed drug classes used in conjunction with conventional therapies for HF include ARNI, SGLT2 inhibitor, HCN channel blocker, sGC stimulator/activator, and cardiac myosin activator. Of the five drug classes, cGMP signaling is directly involved in the mechanism of action of ARNI and sGC stimulator/activator, highlighting the role of cGMP signaling as a potential therapeutic target in the treatment of HF.

There is currently no specific drug approved for the treatment of HFpEF, although efforts are being made to identify and develop specific therapies. Although SGLT2 inhibitors are reportedly effective against both HFpEF and HFrEF, such clinical benefits on the heart may be indirect effects. In addition to the drug classes discussed in this review, several potential drug targets and therapeutic strategies are currently under investigation. Clinical trials of emerging drugs that regulate cGMP signaling are also ongoing. Some drugs are undergoing phase IV trials and may be offered to HF patients in the near future. Although it is not mentioned here, drugs targeting the remodeling (e.g., fibrosis) that occurs with the progression of HF may be developed based on the contribution of cGMP signaling to the remodeling process. However, it is important to note that while these advancements have shown promising results in clinical trials, the effectiveness of any treatment may vary depending on several factors, including the severity and type of HF, the presence of comorbidities, genetic factors, and individual response to medications.

## Figures and Tables

**Figure 1 ijms-24-12866-f001:**
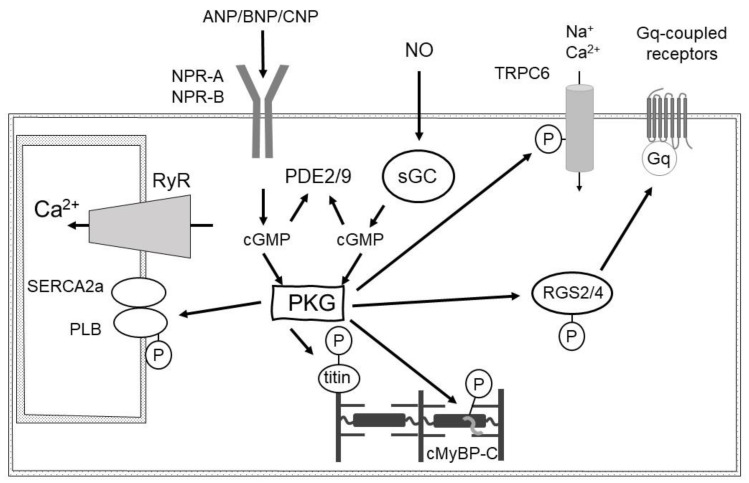
Cyclic guanosine monophosphate (cGMP) signaling in cardiomyocytes. cGMP signaling is initiated by two pathways: NO-sGC and NPRs. cGMP activates PKG, which subsequently phosphorylates various signaling molecules such as RGS2/4 and titin. cGMP also modulates cAMP and cGMP degradation. cGMP-bound PDE2 increases cAMP-hydrolyzing activity. cGMP competitively binds to PDE3, thereby inhibiting its activity. cGMP enhances cAMP action through PDE3 inhibition.

**Figure 2 ijms-24-12866-f002:**
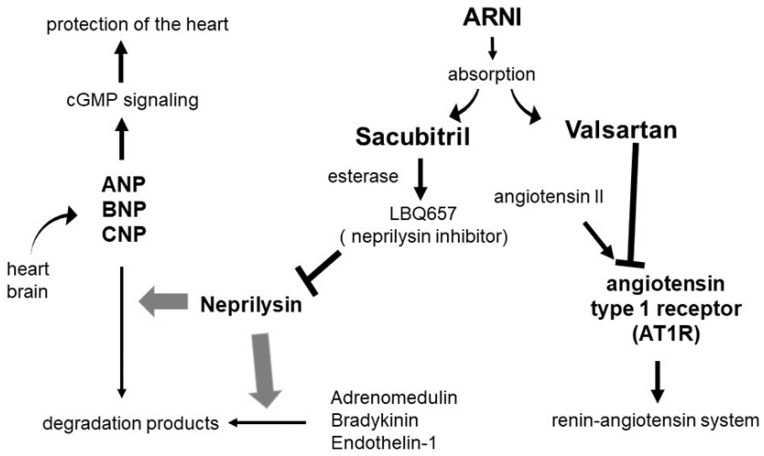
Mechanism of action of angiotensin receptor blocker/neprilysin inhibitor (ARNI). The ARNI drug class consists of two drugs, ARB (valsartan) and the neprilysin inhibitor (sacubitril). Sacubitril is metabolized to LBQ657, which inhibits neprilysin enzymatic activity. Neprilysin hydrolyzes natriuretic peptides (ANP, BNP, and CNP). It also depredates bradykinin and adrenomedullin. Therefore, LBQ657 enhances the effects of endogenous natriuretic peptides.

**Figure 3 ijms-24-12866-f003:**
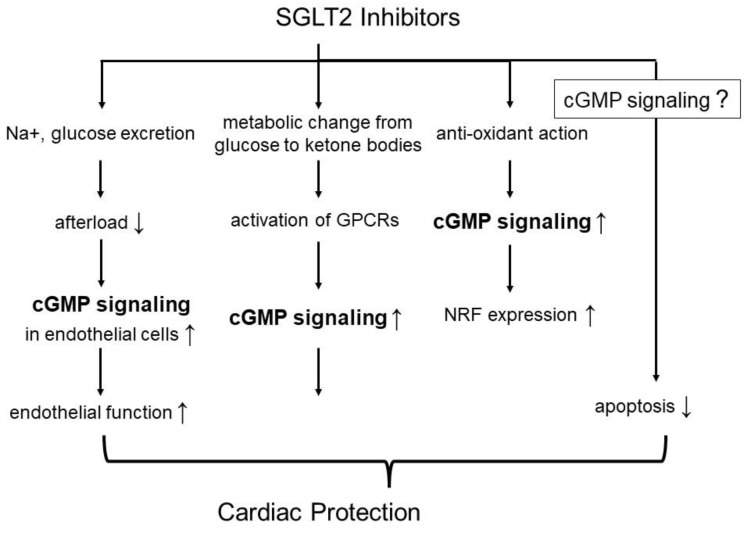
Involvement of cGMP signaling in the cardioprotective mechanism of sodium-glucose co-transporter-2 (SGLT2) inhibitors. Although the contribution of cGMP signaling to cardiac protection has not been established, it is feasible that cGMP signaling contributes to SGLT2 inhibitor-induced cardiac protection. Apoptosis is in part responsible for the protection of the heart. However, there is almost no report on the involvement of cGMP signaling in the inhibition of apoptosis.

**Table 1 ijms-24-12866-t001:** Classification of guanylyl cyclases (GCs).

Type	Molecular Species	Characteristics and Sites of Expression	Ligands and Other Comments
Soluble GC	α1β1 α2β1	Form heterodimers consisting of two types of α (α1 and α2) and two types of β (β1 and β2). β2 does not form a dimer with α1 or α2.	▪ Heterodimers composed of α1 and β1 are physiologically important molecules. ▪ Activated by nitric oxide (NO) binding to sGC ▪ Is deactivated (desensitized) upon S-nitrosylation.
Membrane-bound GC	GC-A (NPR-A)	GC-A and GC-B have natriuretic peptide (NP) binding domains in the amino-terminal region, thus GC-A is referred to as NPR-A, and GC-B is referred to as NPR-B.	▪ Ligands are ANP, BNP, and urodilatin.
	GC-B (NPR-B)		▪ Ligand is CNP.
	NPR-C *	No intracellular GC domain, thus stimulation with NPs does not increase cGMP production.	▪ Involved in the clearance of ANP, BNP, and CNP.
	GC-C	Expressed predominantly in the intestines and partly in the kidneys, liver, and brain.	▪ The ligands for GC-C are gastro-intestinal peptides (guanosine and uroguanosine) and thermostable enterotoxin (STa) produced by *Escherichia coli.*
	GC-D	Pseudogene	-
	GC-E	Expressed in the retina	-
	GC-F	Expressed in the retina	-
	GC-G	Pseudogene	-

* No GC activity.

**Table 2 ijms-24-12866-t002:** Stimulants and activators that act on soluble guanylyl cyclase (sGC).

Effect on sGC Activity	Name of Drug or Compound
Stimulators	Riociguat (BAY 63–2521)
	Vericiguat (BAY 1021189/MK-1242)
	Praliciguat (IW-1973)
	Zagociguat (CY-6463)
	MK-5475
Activators	Runcaciguat (BAY 1101042)
	Mosliciguat (BAY 1237592)
	BI-685509
	Mosliciguat (BAY 1237592)
	BI-685509

**Table 3 ijms-24-12866-t003:** Phosphodiesterase (PDE) family.

PDE Family Members	Selectivity of Cyclic Nucleotides to Hydrolyze	Affinity for cAMP and cGMP	Activity Regulation
PDE1	cAMP, cGMP	PDE1A: cGMP > cAMP PDE1B: cGMP > cAMP PDE1C: cAMP = cGMP	▪ Activity is increased by Ca^2+^/calmodulin.
PDE2	cAMP, cGMP	PDE2A: cAMP = cGMP	▪ Binding of cGMP increases cAMP and cGMP hydrolysis activity.
PDE3	cAMP, cGMP	PDE3A, PDE3B: cAMP = cGMP (catalytic rates: cAMP > cGMP)	▪ Behaves as a substrate that competes with cAMP and cGMP (can be described as a cAMP-hydrolyzing enzyme inhibited by cGMP). ▪ Phosphorylation by PKA increases cAMP hydrolysis activity.
PDE4	cAMP	-	-
PDE5	cGMP	-	▪ Binding of cGMP increases cGMP hydrolytic activity. ▪ Phosphorylation of PKG increases cGMP hydrolytic activity.
PDE6	cGMP	-	▪ Light-activated transducin stimulates PDE6 activity by removing the inhibitory gamma subunit.
PDE7	cAMP	-	-
PDE8	cAMP	-	-
PDE9	cGMP	-	▪ PDE9A degrades cGMP produced by NP-NPR but not cGMP produced by NO-sGC.
PDE10	cAMP, cGMP	PDE10A: cAMP > cGMP	-
PDE11	cAMP, cGMP	PDE11A: cAMP = cGMP	-

**Table 4 ijms-24-12866-t004:** Classification of glucose transporters (GLUTs).

Classification	Subfamily	Isoforms	Characteristics
GLUTs	class I	GLUT1 GLUT2 GLUT3 GLUT4 GLUT14	▪ Within Class I, GLUT4 promotes insulin-stimulated glucose uptake. ▪ In the heart, GLUT4 is involved in glucose uptake.
	Class II	GLUT5 GLUT7 GLUT9 GLUT11	-
	class III	GLUT6 GLUT8 GLUT10 GLUT12 HMIT	▪ Within Class III, GLUT8 and GLUT12 promote insulin-stimulated glucose uptake. ▪ HMIT acts on myoinositol uptake.

**Table 5 ijms-24-12866-t005:** Classification of sodium-glucose cotransporters (SGLTs).

Classification	Isoform	Physiological Functions
SGLTs	SGLT1 SGLT2 SGLT3 SGLT4 SGLT5 SGLT6 (SMIT2 *) SMIT1 *	▪ Myocardium expresses SGLT1 and SMIT1 (glucose taken up by SGLT1 and SMIT1 is used for ROS production but not for ATP production). ▪ SGLT2 inhibitors are effective against HF. ▪ SMIT is involved in the transport of myoinositol but not glucose.

* SMIT: sodium myo-inositol cotransporter.

**Table 6 ijms-24-12866-t006:** Clinical studies of ARNI.

Drug	Study Population	Treatment	Primary and Secondary Endpoints	Main Findings and Conclusions
LCZ696 (PARAMOUNT trial) [112]	▪ HFpEF patients with LVEF ≥ 45% and NYHA II–III. ▪ NT-proBNP level > 400 pg/mL (N = 301).	LCZ696 200 mg BID or valsartan 160 mg BID for 36 weeks	Primary: ▪ Changes in NT-proBNP levels Secondary: ▪ Change in echocardiographic measures. ▪ Change in BP. ▪ Change in NYHA, clinical composite assessment, and QoL.	▪ At 12 weeks, LCZ696 showed greater effects in reducing NT-proBNP levels. ▪ At 36 weeks, LCZ696 was associated with left atrial reverse remodeling and improvement in NYHA functional class. ▪ LCZ696 was well tolerated.
LCZ696 (PARADIGM-HF trial) [113]	▪ HFrEF patients with LVEF ≤ 40% and NYHA II–IV. ▪ NT-proBNP level ≥ 600 pg/mL (N = 8399).	LCZ696 200 mg BID or enalapril 10 mg BID	Primary: ▪ A composite of death from CV causes or a first hospitalization with HF. Secondary: ▪ Time to death from any cause. ▪ Change in the clinical summary score on the KCCQ. ▪ Time to a new onset of atrial fibrillation. ▪ Time to first occurrence of a decline in renal function.	▪ LCZ696 reduced the risks of death and hospitalization from HF. ▪ LCZ696 had a higher occurrence of hypotension and non-serious angioedema compared with enalapril. ▪ LCZ696 had a lower occurrence of renal impairment, hyperkalemia, and cough compared with enalapril.
Sacubitril-valsartan (PARAGON-HF trial) [114]	▪ HFpEF patients with LVEF ≥ 45% and NYHA II–III. ▪ Elevation of NT-proBNP level. ▪ Structural heart disease (N = 4822).	Sacubitril-valsartan 200 mg BID or valsartan 160 mg BID	Primary: ▪ A composite of total hospitalizations due to HF and death from CV causes. Secondary: ▪ Change in the clinical summary score on the KCCQ. ▪ Change in NYHA class. ▪ First occurrence of a decline in renal function. ▪ Death from any cause.	▪ Sacubitril-valsartan did not meet the primary endpoint. ▪ Sacubitril-valsartan had a higher occurrence of hypotension and angioedema. ▪ Sacubitril-valsartan had a lower incidence of hyperkalemia.

**Table 7 ijms-24-12866-t007:** Clinical studies of HCN channel blocker.

Drug	Study Population	Treatment	Primary and Secondary Endpoints	Main Findings and Conclusions
Ivabradine (BEAUTIFUL trial) [117]	▪ Patients with CAD and LV systolic dysfunction ▪ LVEF < 40% ▪ Resting HR ≥ 60 bpm (N = 10,917)	Ivabradine 5–7.5 mg BID or placebo	Primary: ▪ A composite of CV death, admission to hospital for acute MI, and admission to hospital for new-onset or worsening HF.	▪ Ivabradine reduced HR of 6 and 5 bpm at 12 and 24 months, respectively. ▪ Reduction in HR could be used to reduce the incidence of CAD outcomes in patients who have HR ≥ 70 bpm.
Ivabradine (Echo substudy of BEAUTIFUL) [118]	Subgroup analysis of the BEAUTIFUL trial (N *=* 590)		Primary: ▪ Change in LV end-systolic volume index (LVESVI). Secondary: ▪ Changes in LVEF, LVEDVI, and NT-proBNP.	▪ Ivabradine was associated with a decrease in LVESVI and an increase in LVEF. ▪ Ivabradine may reverse detrimental LV remodeling.
Ivabradine (SHIFT trial) [119]	▪ Patients with symptomatic HF and LVEF ≤ 35% ▪ HR ≥ 70 bpm (N = 6558)	Ivabradine titrated to a maximum of 7.5 mg BID or placebo	Primary: ▪ A composite of CV death or hospital admission for worsening HF.	▪ Ivabradine reduced major risks associated with HF. ▪ Ivabradine reduced the risk of the primary endpoint. ▪ No difference in reducing CV and all-cause deaths.
Ivabradine (Echo substudy of SHIFT) [120]	Subgroup analysis of the SHIFT trial (N *=* 411)		Primary: ▪ Change in LVESVI from baseline to 8 months. Secondary: ▪ Changes in LVEF, LVEDVI, LVESV, and LVEDV	▪ Ivabradine reduced LVESVI. ▪ Ivabradine reverses cardiac remodeling in HF patients with LV systolic dysfunction.

**Table 9 ijms-24-12866-t009:** Clinical studies of SGLT2 inhibitors.

Drug	Study Population	Treatment	Primary and Secondary Endpoints	Main Findings and Conclusions
Canagliflozin (CHIEF-HF trial) [133]	▪ HF patients regardless of EF or diabetes status (N = 476).	Canagliflozin 100 mg OD or placebo for 12 weeks.	Primary: ▪ Change in KCCQ TSS from baseline to week 12.	▪ The 12-week change in KCCQ TSS was higher with canagliflozin. ▪ Canagliflozin improved symptom burden in HF patients, regardless of EF or diabetes.
Dapagliflozin (DELIVER trial) [51]	▪ Patients with LVEF ≥ 40% ▪ With or without T2DM ▪ Elevation of natriuretic peptide levels ▪ Structural heart disease (N = 6232)	Dapagliflozin 10 mg OD or placebo	Primary: ▪ A composite of worsening HF or CV death. Secondary: ▪ Total number of worsening HF events and CV death. ▪ Change in KCCQ TSS from baseline.	▪ Total events and symptom burden were lower with dapagliflozin. ▪ Dapagliflozin was associated with greater reductions in the combined risk of worsening HF or CV death in patients with mildly reduced or preserved EF.
Dapagliflozin (DAPA-HF trial) [134]	▪ Patients with LVEF ≤ 40% and NYHA II–IV ▪ NT-proBNP level ≥ 600 pg/mL ▪ With or without T2DM (N = 4744)	Dapagliflozin 10 mg OD or placebo	Primary: ▪ A composite of worsening HF or CV death Secondary: ▪ A composite of hospitalization with HF or CV death. ▪ Total number of CV death and hospitalizations for HF.	▪ Dapagliflozin reduced the risk of worsening HF or death from CV causes, regardless of diabetes. ▪ Dapagliflozin had better symptom scores. ▪ No significant differences in the incidences of adverse events (e.g., hypoglycemia, volume depletion, and renal dysfunction).
Dapagliflozin (DEFINE-HF trial) [135]	▪ Patients with LVEF ≤ 40% and NYHA II–III ▪ Elevation of NT-proBNP or BNP level ▪ eGFR ≥ 30 mL/min/1.73 m2 (N = 263)	Dapagliflozin 10 mg OD or placebo for 12 weeks	Primary: ▪ Average of mean NT-proBNP at 6 and 12 weeks ▪ A composite of the proportion of patients who achieved meaningful improvement in health status.	▪ Dapagliflozin increased the proportion of patients experiencing clinically meaningful improvements in HF-related health status or natriuretic peptides. ▪ No significant differences in the average 6- and 12-week NT-proBNP levels.
Empagliflozin (EMPEROR-Reduced trial) [136]	▪ Patients with LVEF ≤ 40% and NYHA II–IV ▪ Elevation of NT-proBNP level (N = 3730)	Empagliflozin 10 mg OD or placebo	Primary: ▪ A composite of CV death or hospitalization with HF. Secondary: ▪ Occurrence of all hospitalizations for HF. ▪ Rate of decline of eGFR.	▪ Empagliflozin had a lower total number of hospitalizations for HF compared with the placebo, regardless of diabetes status. ▪ Empagliflozin had a slower rate of decline of eGFR and a lower risk of serious renal outcome. ▪ Empagliflozin had a higher incidence of uncomplicated genital tract infection.
Empagliflozin (EMPEROR-Preserved trial) [137]	▪ Patients with LVEF > 40% and NYHA II–IV ▪ NT-proBNP > 300 pg/mL (N = 5988)	Empagliflozin 10 mg OD or placebo	Primary: ▪ A composite of CV death or hospitalization with HF. Secondary: ▪ Occurrence of all hospitalizations for HF.	▪ Empagliflozin reduced the risk of CV death or hospitalization with HF, regardless of diabetes status. ▪ Empagliflozin had a higher incidence of uncomplicated genital and urinary tract infections and hypotension.

**Table 11 ijms-24-12866-t011:** Clinical studies of β_3_AR agonist.

Drug	Study Population	Treatment	Primary and Secondary Endpoints	Main Findings and Conclusion
Mirabegron (BEAT-HF trial) [143]	▪ HF patients with NYHA II–III and LVEF < 40% (N = 70).	Mirabegron titrated to 150 mg BID or placebo for 6 months	Primary: ▪ Change in LVEF from baseline to 6 months.	▪ The primary endpoint of the study was not met. ▪ From exploratory analysis, mirabegron was associated with increased LVEF in patients with severe HF.
Mirabegron (BEAT-HF-II trial) [90]	▪ HF patients with NYHA III–IV and LVEF < 35%. ▪ Increased NT-proBNP levels (N = 22).	Mirabegron 300 mg daily or placebo for one week	▪ Invasive hemodynamic measurements. ▪ Changes in cardiac index, SV index, HR, systemic vascular resistance (SVR), BP, and renal function.	▪ Mirabegron was associated with increased cardiac index and decreased pulmonary vascular resistance. ▪ No significant differences in changes in HR, SVR, BP, or renal function.

## Data Availability

All data generated or analyzed during the current study are included in this published article.

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
