# Peer review of "New Therapeutics for Heart Failure: Focusing on cGMP Signaling"

_ijms, 2023, doi:10.3390/ijms241612866_

Round 1
Reviewer 1 Report
The review “New Therapeutics for Heart Failure: Focusing on cGMP Signaling” is well-written with a lot of useful information. The authors have prepared a comprehensive review of different classes of drugs for heart failure and their clinical studies and covered almost all aspects of the concept. An effort made by the authors for making a separate table for each class of drugs which analyses their clinical studies, mechanism, and outcomes is highly appreciable. Nevertheless, some issues need to address in the manuscript before considering it for publication. Below are some comments that require the author’s attention-
1.. In various sentences I found that the authors have written the classes of drugs however, they have inserted drug names between classifications. For instance, in conclusion, “Of the five new drugs (ARNI, SGLT2 inhibitor, ivabradine, cardiac myosin activator, and sGC stimulator (activator)), cGMP signaling is directly involved in the mechanism of action of ARNI and sGC stimulator/activator”. I would suggest the authors mention either drug names or classes.
3. The authors are advised to maintain the text alignment in the entire manuscript. For instance, table.
4. In the SGLT2 inhibitors section, although the authors have done a great job to summarize the information but there is no deeper critical assessment of data about cGMP signaling in HF since the title of the manuscript is about new therapeutics focusing on cGMP signaling in HF. Authors are suggested to give some information for the same. The authors can also add some figures related to the same.
5. I would appreciate if author can add few lines on limitation of current article.
6. Future recommendations pertaining to cGMP signaling in HF should be added to understand the key gaps and challenges in their therapeutic utility in clinical practice.
Author Response
Reviewer’s comments
Reviewer #1
The review “New Therapeutics for Heart Failure: Focusing on cGMP Signaling” is well-written with a lot of useful information. The authors have prepared a comprehensive review of different classes of drugs for heart failure and their clinical studies and covered almost all aspects of the concept. An effort made by the authors for making a separate table for each class of drugs which analyses their clinical studies, mechanism, and outcomes is highly appreciable. Nevertheless, some issues need to address in the manuscript before considering it for publication. Below are some comments that require the author’s attention-
We would like to thank the reviewer for the insightful comments. We have addressed the concerns raised by the reviewer by adding explanations to each query. Our point-by-point response is presented below, and we have highlighted (shown in blue color) where modifications to the manuscript text have been made.
- In various sentences I found that the authors have written the classes of drugs however, they have inserted drug names between classifications. For instance, in conclusion, “Of the five new drugs (ARNI, SGLT2 inhibitor, ivabradine, cardiac myosin activator, and sGC stimulator (activator)), cGMP signaling is directly involved in the mechanism of action of ARNI and sGC stimulator/activator”. I would suggest the authors mention either drug names or classes.
We would like to thank the reviewer your suggestions.
We have agreed with the reviewer and have revised the drug names/classes in the manuscript.
“angiotensin receptor blocker/neprilysin inhibitor, sodium-glucose co-transporter-2 (SGLT2) inhibitor, hyperpolarization-activated cyclic nucleotide-gated (HCN) channel blocker, soluble guanylyl cyclase stimulator/activator, and cardiac myosin activator” Page 1, line 16-18
“ARNI is the drug class” Page 9, line 299
“Some of the newly developed drug classes used in conjunction with conventional therapies for HF include ARNI, SGLT2 inhibitor, HCN channel blocker, sGC stimulator/activator, and cardiac myosin activator.” Page 32, line 1067-1069
- The authors are advised to maintain the text alignment in the entire manuscript. For instance, table.
We would like to thank the reviewer for pointing out this concern.
We have addressed this issue by adjusting the text alignment in the manuscript especially all tables.
“Revised table 1-11”
- In the SGLT2 inhibitors section, although the authors have done a great job to summarize the information but there is no deeper critical assessment of data about cGMP signaling in HF since the title of the manuscript is about new therapeutics focusing on cGMP signaling in HF. Authors are suggested to give some information for the same. The authors can also add some figures related to the same.
Thank you for your suggestion. We have added the new topic as shown in section 5. Candidate molecular therapeutic targets for HF and cardiac remodeling (Page 16-18)
We have added the mechanisms of SGLT2 inhibitor-mediated cardiac protection, although experiments to confirm the models have not done. New sentences are found from page 14-15. (Section 3.6. SGLT2 inhibitors and cGMP signaling)
- I would appreciate if author can add few lines on limitation of current article.
Thank you for your suggestion. We have added the limitation of current article in the “Conclusions and further directions” section (Page 33)
- Future recommendations pertaining to cGMP signaling in HF should be added to understand the key gaps and challenges in their therapeutic utility in clinical practice
Thank you for your suggestion. We have added the new topic as shown in section 5. Candidate molecular therapeutic targets for HF and cardiac remodeling (Page 16-18)
We also added the further direction and limitation in the “section 7. Conclusions and further directions” (Page 33)

Reviewer 2 Report
In this interesting article, the authors review the molecular pathways on which the drugs for treatment of heart failure are supposed to act. Moreover, they report the data from the major clinical trial on these drugs and their results, as well as the guidelines indications for these drugs' use.
In my opinion this is a well-written and comprehensive review on the subject. my only concern is that for all clinical trials, results should be reported with the statistical significance (p-value as well as OR/HR). The authors reported these values only for the trials on SGLTS-inhibitors, please report them also for all the other trials.
In my opinion the quality of English language is mostly fine and minor editing of are required
Author Response
Reviewer’s comments
Reviewer #2
In this interesting article, the authors review the molecular pathways on which the drugs for treatment of heart failure are supposed to act. Moreover, they report the data from the major clinical trial on these drugs and their results, as well as the guidelines indications for these drugs' use.
We would like to thank the reviewer for the insightful comments. We have addressed the concerns raised by the reviewer by adding explanations to each query. Our point-by-point response is presented below, and we have highlighted (shown in blue color) where modifications to the manuscript text have been made.
In my opinion this is a well-written and comprehensive review on the subject. my only concern is that for all clinical trials, results should be reported with the statistical significance (p-value as well as OR/HR). The authors reported these values only for the trials on SGLTS-inhibitors, please report them also for all the other trials.
We would like to thank the reviewer for your suggestion.
We have agreed with the reviewer and have added the p-value and/or OR/HR in all drug classes as shown by the blue texts in the revised manuscript.

Reviewer 3 Report
Comment:
This paper discusses "New Therapeutics for Heart Failure: Focusing on cGMP Signaling". The main contribution of the paper is "Recently, drugs for the treatment of HF are based on new mechanisms. These drugs are angiotensin receptor blocker/neprilysin inhibitor, sodium-glucose co-transporter-2 (SGLT2) inhibitor, ivabradine, soluble guanylyl cyclase stimulator/activator, and myocardial myosin activator. In the myocardium, cAMP stimulation excites the myocardium to induce responses, whereas cGMP signaling inhibits cAMP-mediated responses. Although all new drugs modulate cGMP signaling, cGMP signaling is a promising therapeutic target. In addition, SGLT2 inhibitors have been shown to be effective not only for HFrEF but also for HF with preserved ejection fraction (HFpEF). Thus, drug treatment is open for HFrEF and HFpEF.".
This is an interesting study and is generally well written and structured. However, in my opinion the paper has some shortcomings in regards to the updated recent references which are recommended and some small sections introducing the adenosine and cannbioids that might modulate the intracellular signaling and then affect inotroic effects
Minor comments:
· Regarding the idea, what about the GPCRs that modulate cGMP or aAMP that might modulate inotropic effects for heart. I suggest the adenosine receptor and possible the cannabinoids receptors (if not involved, mention that though)
· The introduction should be advised to be re-written to be in more logical flow.
· Well written except in some situations. I advise recheck it again.
· The methods in details should be described and analysis as well. Have you followed specific method for gathering research papers?
· Please, suggest future potential experiments in details or future gap of knowledge.
· Please, Specify the most specific protein that you suggest might be related to this topic.
· Please, try to add general paragraph about importance of cannabinoids and adenosine receptors as pharmacological targets for heart failure. (GPCRs targets)
· Please, add figure about the potential future targets such as CB1 and adenosine receptors and connect them to PDE inhibitors
· Although it needs to be in more logical flow, the introduction provides a good, generalized background of the topic. However, why not cite more literature papers.
· Please, add paragraph about drug combinations such as cannabinoid and adenosine receptors ligands……PDE inhibitors
· Table 3 what about the isoforms? Do you have specitic ligands for isoforms?
· Table 7 and 10 is too crowded.
· I think the motivations for this study need to be made clearer
· I recommend make figures to be illustrative.
· The conclusion is too vague
· Have you indicated to potential drug combinations among PDE inhibitors/adenosine receptor agonist/cannabioids….???
· Classify the drugs for systolic or dialstoic HF indications if possible
· Examples of novel and recent references for cannabinoid and adenosine to be cited as a pharmacological targets and potentially for HF :
1. Haddad M, Alsalem M, Saleh T, Jaffal SM, Barakat NA, El-Salem K. Interaction of the synthetic cannabinoid WIN55212 with tramadol on nociceptive thresholds and core body temperature in a chemotherapy-induced peripheral neuropathy pain model. Neuroreport. 2023 May 17;34(8):441-448. doi: 10.1097/WNR.0000000000001910. Epub 2023 Apr 25. PMID: 37096753.
2. Haddad M, Alsalem M, Aldossary SA, Kalbouneh H, Jaffal SM, Alshawabkeh Q, Al Hayek S, Abdelhai O, Barakat NA, El-Salem K. The role of adenosine receptor ligands on inflammatory pain: possible modulation of TRPV1 receptor function. Inflammopharmacology. 2023 Feb;31(1):337-347. doi: 10.1007/s10787-022-01127-3. Epub 2022 Dec 29. PMID: 36580157.
In tables, I suggest to re-consider the writing
Author Response
Reviewer’s comments
Reviewer #3
This is an interesting study and is generally well written and structured. However, in my opinion the paper has some shortcomings in regards to the updated recent references which are recommended and some small sections introducing the adenosine and cannabinoids that might modulate the intracellular signaling and then affect inotropic effects.
We would like to thank the reviewer for the insightful comments. We have addressed the concerns raised by the reviewer by adding explanations to each query. Our point-by-point response is presented below, and we have highlighted (shown in blue color) where modifications to the manuscript text have been made.
Minor comments:
Regarding the idea, what about the GPCRs that modulate cGMP or cAMP that might modulate inotropic effects for heart. I suggest the adenosine receptor and possible the cannabinoids receptors (if not involved, mention that though). / Please, try to add general paragraph about importance of cannabinoids and adenosine receptors as pharmacological targets for heart failure (GPCRs targets). / Please, add figure about the potential future targets such as CB1 and adenosine receptors and connect them to PDE inhibitors. / Please, add paragraph about drug combinations such as cannabinoid and adenosine receptors ligands (PDE inhibitors).
We would like to thank the reviewer for raising this issue.
Several families of GPCRs, including βAR, adenosine receptor, and CB receptors, has been reported to regulate cGMP or cAMP signaling and control contractility of human cardiomyocytes [Salazar et al, 2007]. Cyclic nucleotide PDEs regulate cAMP-mediated signaling by controlling intracellular cAMP content. The cAMP-hydrolyzing activity of several families of PDEs is regulated by cGMP (Zaccolo et al, 2007).
However, in this review, we specifically focused on the role of βAR in cGMP signaling, not on others. βAR is widely recognized as the potential therapeutic target for several types of cardiovascular diseases and modulated by cAMP and cGMP crosstalk signaling. Several cGMP-mediated responses in cardiac cells, including a potentiation of Ca2+ currents and a diminution of the responsiveness to βAR agonist, have been shown to result from the effects of cGMP on cAMP hydrolysis (Zaccolo et al, 2007).
In this review, we comprehensively described the role of βARs in NO signaling in the session 4 “β-adrenergic receptors (βARs) in NO system signaling”. Briefly, βAR is prototypical member of GPCR family which comprising of 3 subtypes, β1, β2 and β3. Although all subtypes mainly couple to Gs, β2AR couples to Gi and β3AR couples to eNOS (possibly via Gi) depending on cellular conditions (Mangmool et al., 2017; Michel et al., 2020). At the cardiomyocytes, stimulation of β1ARs increase the available calcium for contraction and therefore increases cardiac contractility (inotropic effect). However, persistent and prolonged sympathetic stimulation via the β1AR ultimately causes HF (Mangmool et al., 2018).
However, we have added the new topic as shown in “section 5. Candidate molecular therapeutic targets for HF and cardiac remodeling” (e.g., fatty acid receptors, cannabinoids, TRPV1 channel, and aquaporins) (Page 16-18)
References:
Salazar NC, Chen J, Rockman HA. Cardiac GPCRs: GPCR signaling in healthy and failing hearts. Biochim Biophys Acta. 2007;1768(4):1006-18.
Zaccolo M, Movsesian MA. cAMP and cGMP signaling cross-talk: role of phosphodiesterases and implications for cardiac pathophysiology. Circ Res. 2007;100(11):1569-78.
Mangmool, S.; Denkaew, T.; Parichatikanond, W.; Kurose, H. β-Adrenergic receptor and insulin resistance in the heart. Biomol. Ther. 2017, 25, :44-56.
Michel, L.Y.M.; Farah, C.; Balligand, J.L. The beta3 adrenergic receptor in healthy and pathological cardiovascular tissues. Cells 2020, 9, 2584.
Mangmool, S.; Parichatikanond, W.; Kurose, H. Therapeutic targets for treatment of heart failure: Focus on GRKs and β-arrestins affecting βAR signaling. Front. Pharmacol. 2018, 9, 1336.
The introduction should be advised to be re-written to be in more logical flow. Although it needs to be in more logical flow, the introduction provides a good, generalized background of the topic. However, why not cite more literature papers. / I think the motivations for this study need to be made clearer
We would like to thank the reviewer for your suggestion.
We have carefully revised the introduction section of the manuscript to better align with the purpose of our research by focusing on the novel drug classes acted via cGMP signaling as promising therapeutics for HF and also included the relevant references as suggested by the reviewer. (“Introduction”, section)
Well written except in some situations. I advise recheck it again.
We have carefully revised the manuscript.
The methods in details should be described and analysis as well. Have you followed specific method for gathering research papers?
Since this review performed as the narrative review, not as systematic review or meta-analysis, we have briefly included a searching method in the Introduction as follow.
Introduction:
“In this review, we primarily introduce cGMP signaling and then extensively discuss their roles as potential therapeutic agents for HF therapy. The published literatures that reported evidence of novel drug classes for HF acted via cGMP signaling were comprehensively searched from standard electronic databases, such as, PubMed, Embase, ScienceDirect, and Scopus.” (Page 1, Line 65-69)
Please, suggest future potential experiments in details or future gap of knowledge.
We think one of problems is animal model of HFpEF. So far, there is no standard animal model of HFpEF. Therefore, it is very hard to analyze the effects of compounds on HFpEF in vitro. However, we think this is another topic.
Please, Specify the most specific protein that you suggest might be related to this topic.
We think that is protein kinase G. However, we would like not to mention it in the text.
Table 3 what about the isoforms? Do you have specific ligands for isoforms? Table 7 and 10 is too crowded.
We changed the ‘isoforms’ to ’members’ to match the notation of Table (Page 6, Line 196). Selective inhibitors are known for each PDE. However, reference #29 provides an information of selective inhibitors. Thus, we would like not to describe inhibitors. We have carefully edited the table 7 and 10.
I recommend make figures to be illustrative.
The conclusion is too vague.
We would like to thank the reviewer for your suggestion. Figures 1 and 2 seem not to be illustrative. However, the role of each signaling molecule is clearly noted. Thus, we think it is sufficient for our purpose. We added a figure (Figure 3) to explain the mechanisms of SGLT2 inhibitor-mediated cardiac protection, although the involvement of cGMP signaling in the SGLT2 inhibitor-mediated protective actions is not firmly established. It will help to understand the mechanism of SGLT2 inhibitors to protect the heart against heart failure.
We have revised the conclusion section of the manuscript to better align with the purpose of our research as suggested by the reviewer. (“Conclusions and further directions”, Page 33)
Classify the drugs for systolic or diastolic HF indications if possible.
We regret to inform you that the particular point you have raised falls outside the scope of the current review.
Examples of novel and recent references for cannabinoid and adenosine to be cited as a pharmacological targets and potentially for HF.
Thank you for your suggestion. We have added the new topic as shown in section 5 (Page 16-18).
- Candidate molecular therapeutic targets for HF and cardiac remodeling
5.1 Fatty acid receptors
5.2 Cannabinoids
5.3 TRPV1 channels
5.4 Aquaporins
- Haddad M, Alsalem M, Saleh T, Jaffal SM, Barakat NA, El-Salem K. Interaction of the synthetic cannabinoid WIN55212 with tramadol on nociceptive thresholds and core body temperature in a chemotherapy-induced peripheral neuropathy pain model. Neuroreport. 2023 May 17;34(8):441-448. doi: 10.1097/WNR.0000000000001910. Epub 2023 Apr 25. PMID: 37096753.
- Haddad M, Alsalem M, Aldossary SA, Kalbouneh H, Jaffal SM, Alshawabkeh Q, Al Hayek S, Abdelhai O, Barakat NA, El-Salem K. The role of adenosine receptor ligands on inflammatory pain: possible modulation of TRPV1 receptor function. Inflammopharmacology. 2023 Feb;31(1):337-347. doi: 10.1007/s10787-022-01127-3. Epub 2022 Dec 29. PMID: 36580157.

Round 2
Reviewer 1 Report
Authors have addressed all my concerns in the revised version of the manuscript. Thus, I recommend the acceptance of the manuscript in the current form.